# In vivo CRISPR screens reveal SCAF1 and USP15 as drivers of pancreatic cancer

Sebastien Martinez[1], Shifei Wu [1,2], Michael Geuenich [1,2], Ahmad Malik [1,2], Ramona Weber [3], Tristan Woo [4], Amy Zhang[5], Gun Ho Jang[5], Dzana Dervovic[1], Khalid N. Al-Zahrani [1], Ricky Tsai [1], Nassima Fodil [6], Philippe Gros [6], Steven Gallinger[1,5], G. Gregory Neely [7], Faiyaz Notta[4,5], Ataman Sendoel[3], Kieran Campbell [1,2], Ulrich Elling [8] & Daniel Schramek [1,2] ✉

Functionally characterizing the genetic alterations that drive pancreatic cancer is a prerequisite for precision medicine. Here, we perform somatic CRISPR/Cas9 mutagenesis screens to assess the transforming potential of 125 recurrently mutated pancreatic cancer genes, which revealed USP15 and SCAF1 as pancreatic tumor suppressors. Mechanistically, we find that USP15 functions in a haploinsufficient manner and that loss of USP15 or SCAF1 leads to reduced inflammatory TNFα, TGF-β and IL6 responses and increased sensitivity to PARP inhibition and Gemcitabine. Furthermore, we find that loss of SCAF1 leads to the formation of a truncated, inactive USP15 isoform at the expense of full-length USP15, functionally coupling SCAF1 and USP15. Notably, USP15 and SCAF1 alterations are observed in 31% of pancreatic cancer patients. Our results highlight the utility of in vivo CRISPR screens to integrate human cancer genomics and mouse modeling for the discovery of cancer driver genes with potential prognostic and therapeutic implications.

Pancreatic ductal adenocarcinoma (PDAC) is the fourth-leading cause of cancer-related death in industrialized countries and is predicted to be the second-leading cause of cancer death in the United States by 2040[1,2]. Despite recent progress in our understanding of the molecular and genetic basis of this malignancy, 5-year survival rates remain low and do not exceed 10%. PDAC is an epithelial tumor that arises from the cells of the pancreatic duct and represents the vast majority of pancreatic neoplasms. PDAC develops due to the acquisition of cooperating alterations in tumor suppressor and oncogenes as well as chromosomal aberrations, which are thought to either occur gradually by a multi-step process or simultaneously in a single catastrophic event[3,4]. Through these mutational processes, tumors also accumulate hundreds of random bystander mutations, which make it exceedingly hard to interpret genomic data and identify the few real driver mutations that trigger tumor initiation, progression, metastasis, and therapy resistance. Whole exome sequencing studies identified several frequent mutations altering the function of key oncogenes and tumor suppressors such as *KRAS* (93%), *TP53* (72%), *CDKN2A* (44%), *SMAD4* (40%), *RNF43* (8%) and *FBXW7* (5%)[5–7].

In the clinic, genomic technologies are reaching the point of detecting genetic variations with high accuracy in patients. This holds the promise to fundamentally alter clinical practice by personalizing treatment decisions based on the genetic make-up of an individual tumor, commonly referred to as precision medicine. These genomic

[1]Centre for Molecular and Systems Biology, Lunenfeld-Tanenbaum Research Institute, Mount Sinai Hospital, Toronto, ON, Canada. [2]Department of Molecular Genetics, University of Toronto, Toronto, ON, Canada. [3]Institute for Regenerative Medicine (IREM), University of Zurich, Zurich, Switzerland. [4]Princess Margaret Cancer Centre, University Health Network, Toronto, ON, Canada. [5]PanCuRx Translational Research Initiative, Ontario Institute for Cancer Research, Toronto, ON, Canada. [6]Department of Biochemistry, Dahdaleh Institute of Genomic Medicine, McGill University, Montreal, QC, Canada. [7]Dr. John and Anne Chong Lab for Functional Genomics, Charles Perkins Centre, and School of Life and Environmental Sciences, The University of Sydney, Camperdown NSW 2006, Australia. [8]Institute of Molecular Biotechnology of the Austrian Academy of Science (IMBA), Dr. Bohr-Gasse 3, Vienna BioCenter (VBC), 1030 Vienna, Austria. ✉e-mail: schramek@lunenfeld.ca

advances have validated previous findings regarding the most commonly mutated PDAC genes but also led to the identification of a long-tail of recurrent but less frequent alterations in hundreds of genes[8,9]. Multiple hypotheses have been proposed to explain the low frequency and high diversity of those infrequently mutated genes. Some of those mutations might provide a functional alteration similar to that of a major driver. Some long-tail mutations may as well be highly penetrant but simply affect genes that are rarely mutated[10]. Alternatively, some might affect the same pathway or molecular mechanism and cooperate to promote tumor progression as recently shown by our study of rarely mutated long-tail genes in head and neck cancer converging on inactivation of the NOTCH signaling pathway[11]. These long-tail genes often lack biological or clinical validation, and their contribution to PDAC development remains unknown. As such, establishing reliable and genetically traceable in vivo screening platforms to systematically identify putative PDAC driver genes, is a prerequisite to fulfill the promise of precision medicine.

Genetically engineered mouse models (GEMM) constitute the gold standard for genetic perturbation studies. Mouse models of human cancers have provided invaluable insights into the genes and molecular mechanisms that drive cancer development[12,13] and proven essential as a preclinical model in the development of novel therapeutic agents[14]. However, conventional GEMMs are extremely time and resource-intensive rendering them impractical to sift through the scores of genetic alterations emerging from large-scale genomics projects[15].

Here, we report a direct in vivo CRISPR/Cas9 screening strategy to identify which long-tail PDAC genes and associated pathways cooperate with oncogenic *Kras*[G12D] to accelerate pancreatic cancer progression and identify *USP15* or *SCAF1* as pancreatic tumor suppressors that regulate inflammatory responses and sensitivity to PARP inhibition and Gemcitabine.

## Results

### Direct in vivo CRISPR gene editing in the mouse pancreas

To functionally test putative PDAC cancer genes in vivo, we employed a multiplexed CRISPR/Cas9 genome editing approach to generate knock-out clones directly in the pancreatic epithelium of tumor-prone mice. We used conditional Lox-Stop-Lox-(LSL)-*Kras*[G12D] and LSL-*Cas9-GFP* mice crossed to the pancreas-specific PDX1-Cre driver line (termed KC mice) and injected an adeno-associated virus that expresses a sgRNA and the H2B-RFP fluorescent marker (AAV-sgRNA-RFP) (Fig. 1a). Cre-mediated excision of Lox-Stop-Lox cassettes resulted in expression of oncogenic *Kras*[G12D], *Cas9* and *GFP* and formation of hundreds of cytokeratin19 positive (CK19) pancreatic intraepithelial neoplasia (PanIN) precursor lesions, which can be lineage-traced by virtue of red fluorescence (Supplementary Fig. 1a, b). To validate the efficiency of CRISPR/Cas9-mediated mutagenesis, we injected sgRNAs targeting *GFP*, which revealed a knock-out efficacy of $78 \pm 6\%$ (Supplementary Fig. 1c).

KC mice exhibited rapid growth of pre-invasive PanINs precursor lesions but showed very slow progression to invasive PDAC with a median latency of 14 months (Fig. 1b). Additional genetic alterations such as loss of transformation-related protein 53 (*Trp53*), *p16Ink4a*, *Lkb1* or inactivation of TGF-β signaling was previously shown to cooperate with *Kras*[G12D] and induces rapid PDAC development within 3-5 month[16–20]. To test whether our direct in vivo CRISPR approach can reveal genetic interactions, we recapitulated cooperation between oncogenic *Kras*[G12D] and loss of p53 (*Trp53*). Indeed, Cas9-mediated ablation of *Trp53* in KC mice triggered rapid PDAC formation with a median latency of 14 weeks, while littermates transduced with scrambled control sgRNAs remained cancer-free for over 1 year (Supplementary Fig. 1d). This is in line with previous efforts using CRISPR/Cas9 gene editing in *KRas*[G12D] mice[21,22] and demonstrates that this approach can be used to test for genetic cooperation between PDAC genes.

### CRISPR Screen identifies Usp15 and Scaf1 as PDAC tumor suppressors

In pancreatic cancer, 125 genes show recurrent somatic mutations[6,7]. To assess these genes in vivo, we established a sgRNA library targeting the corresponding mouse orthologs (4 sgRNAs/gene; 500 sgRNAs) and a library of 420 non-targeting control sgRNAs (Supplementary Data 1). Of note, we did not include sgRNAs targeting well-established PDAC driver genes such as *Trp53* or *Smad4*[16–20].

Next, we optimized the parameters for an in vivo CRISPR screen. Using a mixture of AAV-GFP and AAV-RFP, we determined the viral titer that transduces the pancreatic epithelium at clonal density (MOI < 1). Higher viral titers were associated with double infections, whereas a 15% overall transduction level minimized double infections while generating necessary clones sufficient to screen (Supplementary Fig. 1e). Using multicolor Rosa26-Lox-Stop-Lox(R26-LSL)-Confetti Cre-reporter mice, we next determined the viral titer required to generate thousands of discrete clones within the pancreatic epithelium (Supplementary Fig. 1f). Thus, at a transduction level of 15% and a pool of 500 sgRNAs, each sgRNA would be introduced into at least 50 CK19+ epithelial cells within a single pancreas.

To uncover long-tail genes that cooperate with oncogenic *Kras*[G12D] and accelerate PDAC development, we injected the experimental and the control AAV-sgRNA libraries into the pancreas of 23 and 13 KC mice, respectively. Next-generation sequencing confirmed efficient AAV transduction of all sgRNAs (Supplementary Fig. 2a). Importantly, KC mice transduced with the long-tail PDAC sgRNA library developed pancreatic cancer significantly faster than littermates transduced with the control sgRNA library (31 versus 59 weeks; p < 0.0001) (Fig. 1b and c). In addition, 13/23 (56%) KC mice transduced with the long-tail PDAC sgRNA library developed liver and/or lung metastasis, while only 1/13 ( ~ 8%) littermate mice transduced with the control sgRNA library developed metastasis (Supplementary Fig. 2b-e), indicating the existence of strong tumor suppressors within the long-tail of PDAC associated genes.

To identify these PDAC driver genes, we examined the sgRNA representation in 151 tumors. 78% of tumors showed strong enrichment for single sgRNAs, indicating a clonal origin. In contrast, the pancreas of control-transduced mice with multifocal PanINs showed enrichment of several non-template control sgRNAs (Fig. 1d). We prioritized genes that were targeted by ≥2 sgRNAs and knocked out in multiple tumors and/or metastatic foci, resulting in 8 candidate tumor suppressor genes (Fig. 1e, Supplementary Data 2). These candidates included well-known PDAC tumor suppressor genes, such as *Cdkn2a*[23], *Rnf43*[24], *Fbxw7*[25] or *NF2*[26], as well as genes with poorly understood function, such as *Usp15* and *Scaf1*.

Pancreatitis is a risk factor for the development of PDAC in humans and cooperates with oncogenic *KRas* mutations to induce PDAC formation in mice[23,27]. Therefore, we repeated our screen and treated mice with chronic, low doses of cerulein to induce mild pancreatitis. As expected, cerulein treatment significantly accelerated PDAC development in KC mice transduced with the PDAC sgRNA library (17 versus 32 weeks median survival, *p* < 0.0001), and a trend towards faster PDAC development in KC mice transduced with the control library (Supplementary Fig. 2f). In line with the previous screen, *Cdkn2a* was the top-scoring gene followed by *Rnf43* and the newly identified genes, *Usp15* and *Scaf1* (Supplementary Fig. 2g), further supporting their function as strong suppressors of pancreatic cancer in KC mice.

### Usp15 is a haploinsufficient PDAC tumor suppressor regulating TGFβ, WNT, and NFκb signaling

The multi-domain deubiquitinase USP15 regulates diverse processes, such as the p53 tumor suppressor pathway[28], MAPK signaling[29], Wnt/beta-catenin signaling[30], TGF-β signaling[31–33], NfKb signaling[32,34,35] and chromosome integrity[36,37], either through regulated de-ubiquitination of direct substrates such as MDM2, APC, SMADs or TGF-β receptors, or de-ubiquitination-independent functions such as through protein-protein interactions[38].

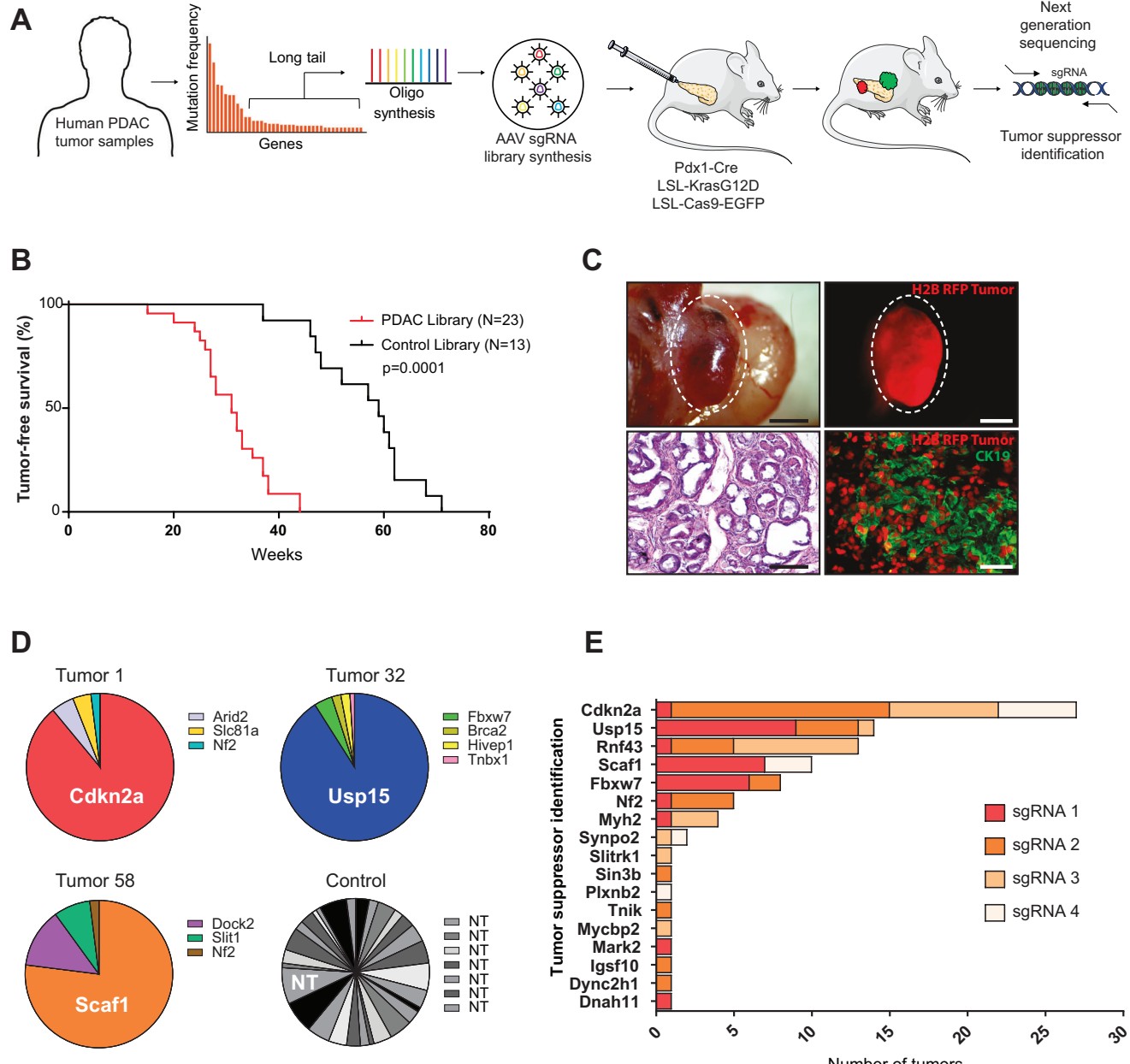

**Fig. 1 | In vivo CRISPR screen reveals pancreatic cancer tumors suppressors.**
**A** Experimental design of the in vivo PDAC CRISPR screen, showing gene selection from long-tail mutations, pancreatic injection of AAV libraries and tumor sequencing. **B** Tumor-free survival of Pdx1-Cre;LSL-*Kras*[G12D];LSL-*Cas9-GFP* mice transduced with a sgRNA library targeting putative pancreatic cancer genes (*n* = 23) or a control sgRNA library (*n* = 13) **C** Representative whole-mount, H&E and immunofluorescent images of an H2B-RFP+ pancreatic PDAC-library tumor: Scale bar 2 mm. H&E image:

scale bar 250 μm. Representative immunofluorescence image shows H2B-RFP and CK19 expression. Scale bar 50 μm. Similar results were observed in all collected tumor. **D** Representative pie charts showing tumor suppressor genes with enriched sgRNAs in tumor DNA obtained from three different pancreatic tumors and a control-transduced pancreas with multifocal PanINs. **E** Bar graph showing putative tumor suppressor genes with enriched sgRNAs in tumor DNA obtained from the PDAC mouse model (sgRNA enriched per tumors are indicated by color).

To validate the tumor suppressive function of Usp15, we first injected KC mice individually with one library or one newly designed sgRNA. All transduced mice developed highly proliferative pancreatic tumors with much shorter latencies compared to mice transduced with the non-targeting control sgRNAs (sgCrtl) (Fig. 2a). In fact, age-matched control KC mice only exhibited PanINs at the time when USP15 knockout mice exhibit aggressive PDACs (Fig. 2b). All tested tumors exhibited efficient CRISRP/Cas9-mediated mutagenesis of *Usp15* (Supplementary Fig. 3a and b).

To further confirm the tumor suppressive role and rule out any confounding effect of Cas9 endonuclease expression, we generated conditional *Usp15*[fl/fl]; *KRas*[G12D]; Pdx1-Cre. This conventional knock-out

approach recapitulated our CRISPR/Cas9 findings (Fig. 2c and d), validating our in vivo CRISPR approach. Interestingly, *Usp15*[fl/+] heterozygous mice also manifested significantly shorter disease-free survival (Fig. 2c). To assess whether tumor development was due to *Usp15* loss of heterozygosity, we used fluorescence-activated cell sorting (FACS) to isolate tumor cells from *Usp15* homozygous, heterozygous and wild-type *KRas*[G12D] tumors. Western Blot analysis revealed Usp15 expression in *Usp15* heterozygous tumor cells, albeit at a reduced level compared to control tumors (Supplementary Fig. 3c), indicating *Usp15* functions as a haploinsufficient tumor suppressor.

Next, we established primary PDAC cell lines from KC mice as well as KC mice with concomitant expression of the hotspot p53[R270H]

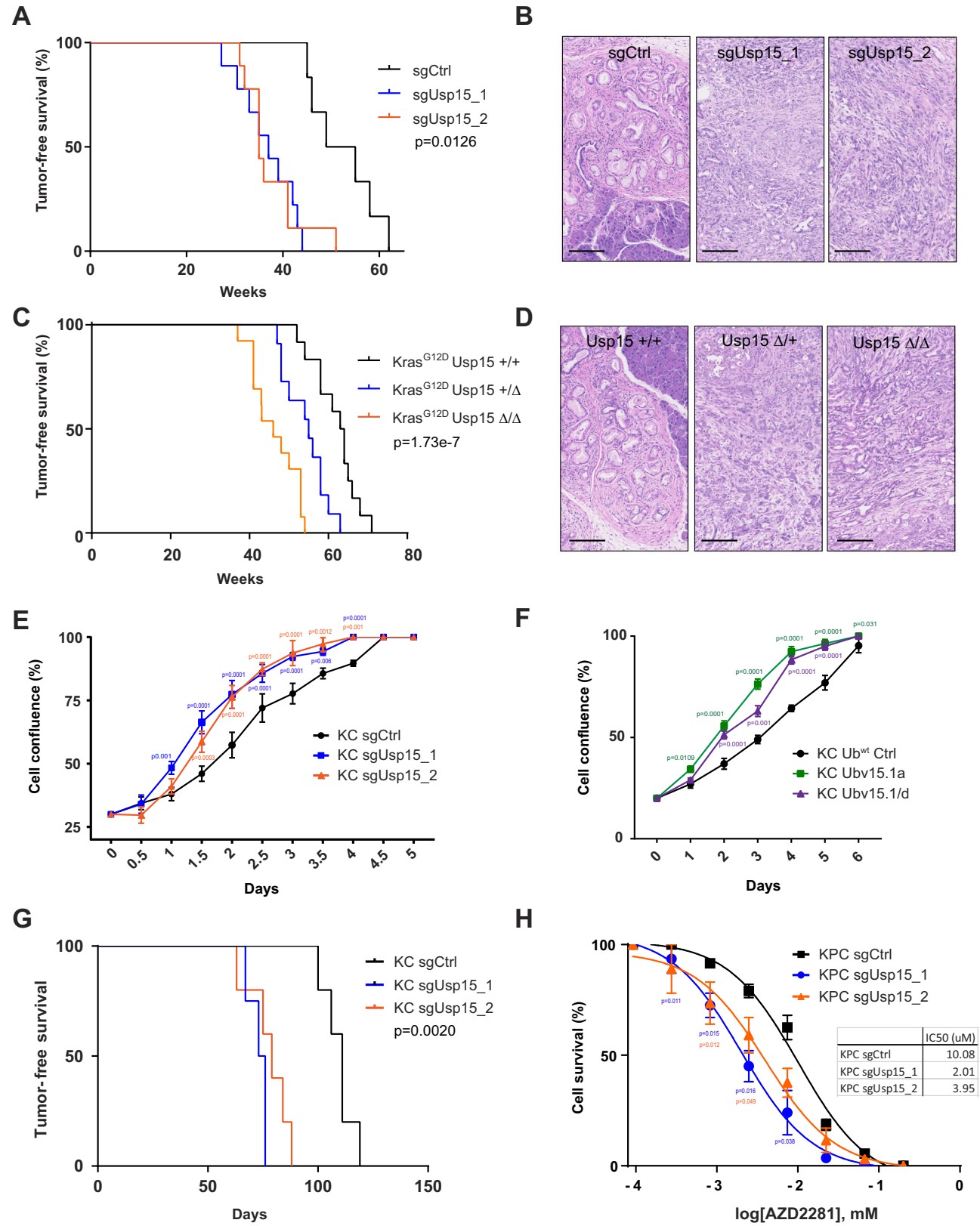

mutant (KPC) and used CRISPR/Cas9 to knock-out *Usp15* (Supplementary Fig. 3d). Loss of *Usp15* significantly increased proliferation of these KC cells (Fig. 2e), while it did not affect KPC cells (Supplementary Fig. 3e), presumably, because those cells are at the maximal proliferation rate. Similar results were obtained using ubiquitin variants (UbVs) that bind and block the catalytic domain of Usp15[38], indicating that this tumor suppressive function is de-ubiquitination dependent

(Fig. 2f). Upon orthotopic injection, *Usp15* knock-out KC cells also formed allograft tumor faster than non-targeting control cells (Fig. 2g). Together, these data show that Usp15 regulates tumor cell proliferation in a cell-autonomous manner and loss of *Usp15* increases a cell's ability to form allograft tumors.

Consistent with a previous report[36], we also found that loss of *Usp15* sensitizes pancreatic cancer cells to Poly-(ADP-ribose)

**Fig. 2 | Usp15 functions as PDAC tumor suppressor. A** Tumor-free survival of Pdx1-Cre;LSL-*Kras*^G12D;LSL-*Cas9-GFP* mice injected with CRISPR AAV-sgRNAs targeting the indicated gene or non-targeting control sgRNA (sgCtrl, *n* = 6). Two independent sgRNAs were used (sgUsp15_1 *n* = 9, sgUsp15_2 *n* = 9). Log-Rank test (Mantel-Cox). **B** Representative H&E images showing multifocal PanINs in sgCtrl transduced pancreas and PADC tumors in sgUsp15 transduced pancreas. Scale bar 100 μm. **C** Tumor-free survival of Pdx1-Cre;LSL-*Kras*^G12D mice with the indicated *Usp15* genotype where '+' indicates the wildtype allele and 'Δ' indicates a conditionally deleted allele. KrasG12D +/- Usp15 +/+ (*n* = 12); KrasG12D +/- Usp15 +/- (*n* = 11); KrasG12D +/- Usp15 -/- (*n* = 13). Log-Rank test (Mantel-Cox).
**D** Representative H&E images of mice with the indicated genotype showing multifocal PanINs and PADC tumors. Scale bar 100 μm. **E** Cell proliferation curves of KC cells transduced with the indicated sgRNA obtained using the IncuCyte live-cell

imaging. Cells were grown for five days and data are expressed as cell confluence percentage (%; mean ± SD, *n* = 3 independent experiments, Two-way ANOVA (sgUsp15_1 *p* = 3.43e-8: sgUsp15_2 *p* = 4.89e-9), Dunnett's multiple comparison. **F** Cell proliferation curves of KC cells expressing ubiquitin variants inhibiting Usp15 (Ubv15.1a and Ubv15.1/d) or wildtype ubiquitin (Ub^wt) as control. (%; mean ± SD, *n* = 3 independent experiments, Two-way ANOVA (Ubv15.1a *p* = 2.74e-6: Ubv15.1/d *p* = 2.06e-7), Dunnett's multiple comparison. **G** Tumor-free survival of NSG (NOD *Scid Gamma*) mice after orthotopic injection sgCtrl (*n* = 5) or sgUsp15_1/2 (*n* = 5; *n* = 5) KC cells. Two independent sgRNAs were used. Log-Rank test (Mantel-Cox). **H** Dose-response curves for KPC sgCtrl or sgUsp15 cells treated with the indicated concentration of Olaparib (mean ± SD, n = 3 independent experiments). Two-way ANOVA (sgUsp15_1 *p* = 0.0349: sgUsp15_2 *p* = 0.0431), Dunnett's multiple comparison.

polymerase inhibition (PARPi) by Olaparib. This increased drug sensitivity was stronger in KPC cells than KC cells and was also seen in response to Gemcitabine, one of the most commonly used chemotherapies to treat pancreatic cancer (Fig. 2h and Supplementary Fig 3f, Fig. 4a and b). KC cells were overall more sensitive to Gemcitabine likely due to the intact p53 response (Supplementary Fig 3g). Importantly, loss of USP15 also sensitized allograft tumors in vivo towards Olaparib treatment (Supplementary Fig. 4c). In addition, we found that Olaparib and Gemcitabine treatment significantly increases expression of Usp15 in KC and KPC cells (Supplementary Fig. 4d). In line with its haploinsufficient tumorigenic effect, heterozygous loss of *Usp15* also significantly increased proliferation and sensitized to Olaparib treatment, but not as pronounced as complete *Usp15* loss (Supplementary Fig. 4e and f). As such, Usp15 appears to function as a double-edged sword in pancreatic cancer, where the loss of *Usp15* enhances tumor progression in the initial stages of tumorigenesis but sensitizes to certain treatment regimens in the later stages.

Given the wide range of USP15 substrates and USP15-regulated pathways with well-known functions in cancer, we set out to elucidate USP15's exact role in PDAC suppression. First, we transcriptionally profiled primary KC cells transduced with sgRNAs targeting *Usp15* or non-template controls sgRNAs. Inactivation of *Usp15* resulted in dramatic changes in gene expression compared to scrambled control *Kras*^G12D tumor cells (794 differentially expressed genes (DEG), false discovery rate (FDR, Benjamini-Hochberg) < 0.05 and absolute log2 fold-change > 1, Fig. 3a and Supplementary Data 3). Gene set enrichment analyses (GSEA) revealed significantly upregulated gene sets associated with xenobiotic detoxification, glutathione metabolism, anabolic processes, and oxidative phosphorylation (Fig. 3b and Supplementary Data 3). These findings are in line with USP15's known role in negatively regulating NRF2[39] (encoded by the NFE2L2 gene), the master regulator of glutathione metabolism and the redox balance of a cell. In addition, NRF2 expression is induced by oncogenic KRAS and known to stimulate proliferation and suppress senescence of PDAC cells[40]. Indeed, *Usp15* knock-out cells exhibited significantly increased levels of Nrf2 (Supplementary Fig. 5a).

GSEA also revealed depleted genes sets associated with inflammatory responses, TNFα, TGFβ and p53 signaling (Fig. 3b-d and Supplementary Fig. 5b), all pathways with well-known tumor suppressive function in PDAC development[17,41]. Quantitative RT-PCR confirmed reduced expression of TNFα and TGFβ responsive genes at baseline as well as TNFα/TGFβ-stimulated conditions (Fig. 3d and Supplementary Fig. 5c). In addition, loss of USP15 reduced TNFα−induced cell death and TGFβ-induced migration (Supplementary Fig. 5d and e). Together, these data indicate that Usp15 functions as a strong haploinsufficient PDAC tumor suppressor potentially by regulating tumor suppressive cytokine signaling pathway.

## SCAF1 is a PDAC tumor suppressor and regulates USP15 levels
Our second new hit, SCAF1 (SR-Related CTD Associated Factor 1), is a member of the human SR (Ser/Arg-rich) superfamily of pre-mRNA

splicing factors. It interacts with the CTD domain of the RNA polymerase II (RNAPII) and is thought to be involved in pre-mRNA splicing[42]. Its close homologs SCAF4 and SCAF8 were recently shown to be essential for correct polyA site selection and RNAPII transcriptional termination in human cells[43]. SCAF1 was also one of the top-scoring hits in a screen for genes that can restore homologous recombination in *BRCA1*-deficient cells and thus conferred resistance to PARP inhibition[44]. However, the molecular function of SCAF1 remains completely elusive.

First, we validated the tumor suppressive function of Scaf1 by injecting KC mice individually with one library or one newly designed sgRNA. All transduced mice developed highly proliferative pancreatic cancer with much shorter latencies compared to mice transduced with the non-targeting control sgRNAs (Fig. 4a, b). Of note, both *Scaf1* sgRNAs induced high CRISPR/Cas9-mediated mutagenesis and resulted in significantly reduced *Scaf1* mRNA expression (Supplementary Fig. 6a, b). Similar to *Usp15* knockout cells, we also found that primary *Scaf1* knockout KC cells exhibited increased proliferation in culture and formed tumors faster when injected orthotopically into mice compared to scrambled control KC cells (Fig. 4c and d). *Scaf1* knockout cells also exhibited significantly increased sensitivity to Olaparib in vitro and in vivo (Fig. 4e and Supplementary Fig. 6c and d), again phenocopying *Usp15* knockout cells.

Interestingly, we found a connection between Scaf1 and Usp15. *Scaf1* knockout cells exhibited reduced expression of full-length Usp15 (molecular weight of ~125 kDa) and showed expression of a 25 kDa short Usp15 isoform under homeostatic as well as Olaparib and gemcitabine treatment (Fig. 4f and Supplementary Fig. 6e and f). SCAF1 KO tumors also exhibited lower levels of full-length USP15 (Supplementary Data Fig. 6g). To examine a potential function of this truncated isoform, we cloned and transduced the long and the short isoforms into primary Usp15 knock-out KC cells (Supplementary Fig. 6h). While full-length Usp15 was able to supress the hyperproliferative phenotype of Usp15 knock-out cells, the short isoform failed to suppress the cell proliferation (Supplementary Fig. 7a). Similarly, re-expressing the full-length but not the short Usp15 isoform reversed the sensitivity of Usp15 knock-out KC cells to Olaparib and gemcitabine (Supplementary Fig. 7b). In addition, overexpression of the full-length or the short Usp15 isoform did not alter proliferation of wildtype KC cells (Supplementary Fig. 7c), indicating that the short isoform does not exhibit dominant negative functions. However, expression of the long but not the short Usp15 isoform or a catalytically-dead USP15 isoform suppressed the hyperproliferation and Olaparib sensitivity as well as the increased in vivo tumorigenesis of Scaf1 knock-out cells (Fig. 4g and h and Supplementary Fig. 7c-f). Together these data indicate that the short isoform has no tumor suppressive functions or alters the response to PARP inhibition and that Scaf1's tumor suppressive function is at least in part routed by regulating the expression of full-length Usp15.

To further elucidate the effects of Scaf1, we transcriptionally profiled Scaf1 knockout KC cells. Inactivation of *Scaf1* resulted in 625 differentially expressed genes (DEG) (false discovery rate (FDR) < 0.05 and

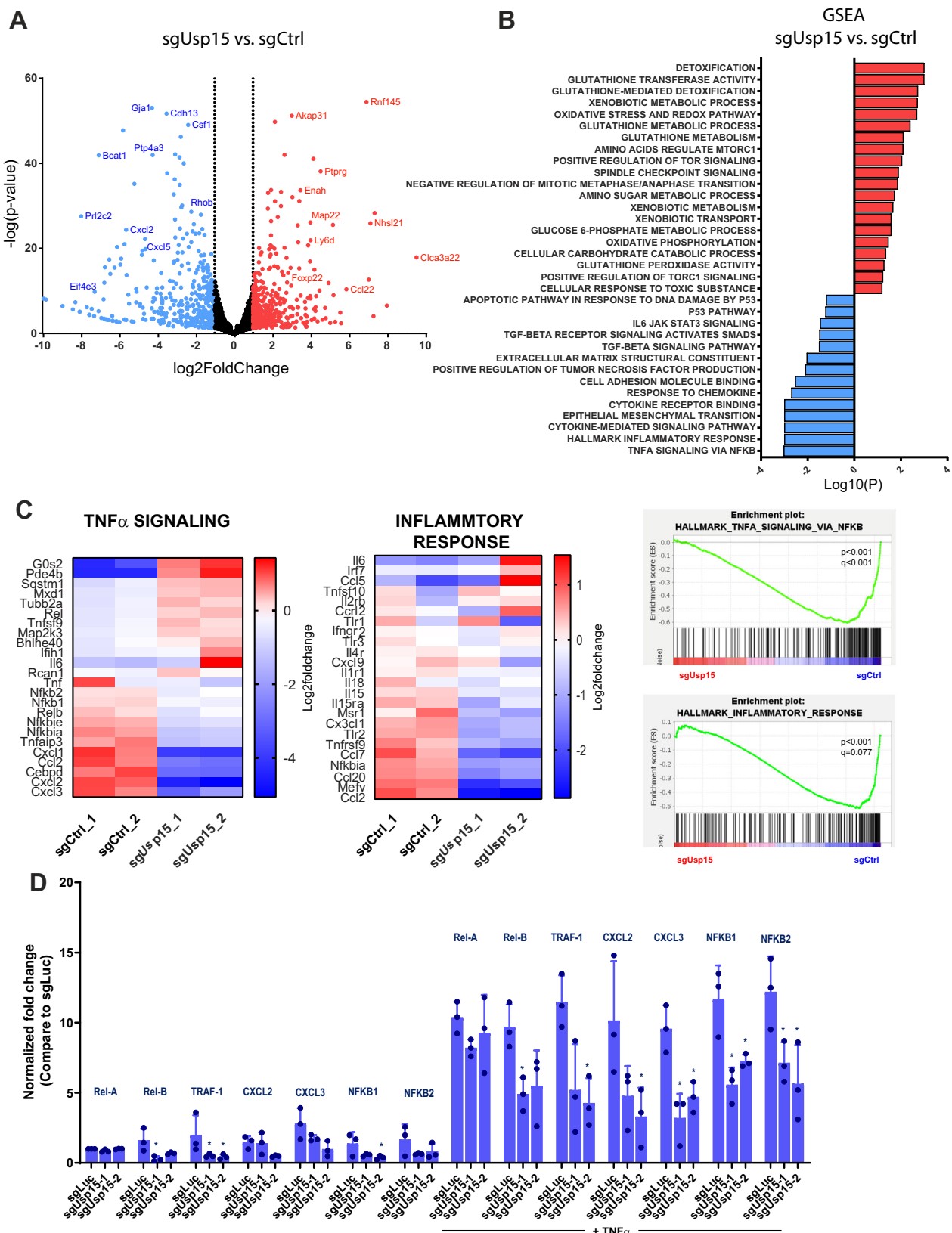

absolute log2 fold-change > 1, Fig. 5a and Supplementary Data 3) compared to scrambled control $Kras^{G12D}$ tumor cells. GSEA revealed significantly upregulated gene sets associated with nucleotide metabolism, glutathione metabolism, microtubule polymerization, and oxidative phosphorylation as well as downregulation gene sets associated with TNFα signaling, one-carbon metabolism, xenobiotic catabolic processes,

mTorc1/mTOR signaling, hypoxia and p53 signaling (Fig. 5b and Supplementary Fig. 7g). In addition, we found a trend towards downregulated TGFβ signaling (Fig. 5b and Supplementary Data 3), reminiscent of the pathways altered in Usp15 knock-out cells.

Lastly, we set out to elucidate how Usp15 and Scaf1 regulate the response of pancreatic cancer cells to PARP inhibition. Interestingly,

**Fig. 3 | Usp15 regulates several pathways involved in PDAC development.**
**A** Volcano Blot showing differential expressed genes between Usp15-knockout compared to sgCtrl control KC cells. Wald test and Benjamini-Hochberg (BH)-adjusted P-value. Two independent sgRNAs, two biological duplicates. **B** Bar graph showing gene set enrichment analysis (GSEA) of Usp15-knockout compared to sgCtrl control KC cells. GSEA nominal p-values. Two independent sgRNAs, two biological duplicates. **C** GSEA plots and Heatmaps of log2 counts per million for selected differentially expressed pathways and genes in sgUsp15 versus sgCtrl control KC cells. GSEA nominal p-values. Two independent sgRNAs, two biological duplicates. **D** Expression levels of genes related to TNFα signaling evaluated by RT-qPCR. Results were normalized with Gapdh and are expressed in fold change compared to Ctrl (mean ± SEM, $n = 3$ independent experiments). Cells were incubated with 10 ng/mL TNFα-for 30 min. Two-sided T-test, Rel-B $p = 0.043$; TRAF-1 $p = 0.037/p = 0.034$; NFKB1 $p = 0.036$; Rel-B $p = 0.042$; TRAF-1 $p = 0.039$; CXCL2 $p = 0.028$; CXCL3 $p = 0.047/p = 0.043$; NFKB1 $p = 0.038/p = 0.040$; NFKB2 $p = 0.039/p = 0.043$.

transcriptional profiling and GSEA following Olaparib treatment revealed that both Usp15 and Scaf1 knock-out cells, exhibited down-regulation of hedgehog signaling, TGFβ signaling and 'axon guidance by netrin' as well as upregulation of 'glycolysis' as the top dysregulated pathways compared to Olaparib-treated control KC cells (Fig. 5c and Supplementary Data 4). Together, this indicates a common mechanism leading to increased sensitivity to PARP inhibition shared between Usp15 and Scaf1 knock-out cells. Indeed, quantitative RT-PCR confirmed reduced expression of hedgehog target genes at baseline as well as upon sonic hedgehog stimulation (Fig. 5d). Thus, Scaf1 and Usp15 knockout cells share several alterations such as upregulated TNFα signaling and downregulated TGFβ, hedgehog and p53 signaling but also several distinct pathways.

### USP15 and SCAF1 in human PDAC

To extend our findings from mouse to human cancers, we analyzed 295 PDAC samples from The Cancer Genome Atlas[45–47]. Mutations and homozygous deletion of *USP15* and *Scaf1* are rare as expected for long-tail mutation and were found in only 2.4% and 1.4% of PDAC samples, respectively. However, an additional 25% and 13% of PDAC cases showed shallow deletions of *USP15 and SCAF1*, respectively, indicative of heterozygous loss of these genes (Fig. 6a). Focal *USP15* and *SCAF1* copy-number losses have been identified in independent large-scale genome studies[48,49]. In addition, allelic copy number loss also coincided with reduced expression of *USP15* and *SCAF1* and patients with deep or shallow *USP15* or *SCAF1* deletions showed a significant trend towards a shorter overall survival (Fig. 6b and Supplementary Fig. 8a). Given our genetic and biochemical data linking SCAF1 and USP15, we next considered patients with deep or shallow *USP15* or *SCAF1* deletions as a group (=37% of patients) and found a significantly shorter overall survival (Supplementary Fig. 8b). This raises the possibility that USP15 and potentially also SCAF1 function in a haploinsufficient manner, which is in line with the increased tumorigenesis found in the *Usp15*$^{fl/+}$; *KRas*$^{G12D}$; Pdx1-Cre mice.

Next, we assessed the expression of USP15 in 4 human pancreatic cancer cell lines. While PANC1 and HPAFII exhibited expression of the small as well as the long USP15 isoform, MiaPACA2 and BXPC3 cells only exhibited low-level expression of the long USP15 isoform, indicating that USP15 is also downregulated in some human pancreatic cancer cell lines (Supplementary Fig. 8c).

To functionally test *USP15* and *SCAF1*, we genetically ablated these genes in human PANC1 cells (Supplementary Fig. 8d and e). Importantly, genetic ablation of *SCAF1* resulted in increased expression of the short USP15 isoform, indicating that this mechanism is conserved from mouse to human cells (Supplementary Fig. 8f). Similarly, to our autochthonous mouse experiments, we also found that loss of *USP15* or *SCAF1* in PANC1 cells resulted in accelerated tumorigenesis and increased sensitivity to Olaparib and Gemcitabine (Fig. 6c, d and Supplementary Fig. 8g). We also observed increased NRF2 protein levels in USP15 knockout PANC1 cells, which showed further elevated upon inhibition of TXNRD1/2 and antioxidant imbalance by auranofin treatment[50] (Supplementary Fig. 8h), akin to our findings in mouse KC cells. USP15 knockout PANC1 cells also exhibited increased sensitivity to auranofin treatment (Supplementary Fig. 8i).

Lastly, we genetically ablated USP15 in patient-derived organoids (PDOs) from 3 different pancreatic cancer patients using Cas9

ribonucleotide particles. We set up competitive growth assays to assess the relative fitness of USP15 knockout PDOs compared to OR2W5 knockout PDOs. Of note, the OR2W5 olfactory receptor is not expressed in pancreatic PDOs and thus serves as control. We mixed the USP15 knockout and the OR2W5 knockout PDOs at a 1:4 ratio and followed their relative growth by quantifying the percent of USP15 and OR2W5 mutations over time using Sanger sequencing. Within ~10 passages, we observed that the PDO cultures were almost completely taken over by USP15 knockout cells (Fig. 6e). Together, these data demonstrate the tumor suppressive function of USP15 and SCAF1 in pancreatic cancer by modulating several important signaling pathways and that loss of USP15 and SCAF1 sensitizes to Gemcitabine and Olaparib.

## Discussion

One key bottleneck on the path toward 'Precision Medicine' is our limited understanding of the functional consequence of most genetic alterations associated with specific malignancies. Cancer develops due to the acquisition of cooperating alterations in tumor suppressor and oncogenes (=driver mutations), which are thought to either occur gradually or simultaneously in a single catastrophic event (e.g. chromothripsis) as recently shown by Notta et al. [4]. Through these mutational processes tumors also accumulate hundreds of random bystander mutations, which make it exceedingly hard to interpret genomic data and identify the few real driver mutations that trigger tumor initiation, progression, metastasis and therapy resistance. Even within known cancer genes, many variants are of uncertain significance (VUS), where the effect of the genetic alteration on gene function cannot be definitely predicted using current bioinformatics tools. Genetic-based treatment design is thus reliant on weeding out bystanders and identifying bona fide driver mutations, as only the latter have diagnostic and therapeutic value. In addition, we have to identify the actionable nodes within a given cancer gene network that can be exploited to selectively kill or disable cancer cells. We also have to identify cancer genotypes that either confer sensitivity or resistance to a given treatment to be able to stratify patients into the best treatment arm. Lastly, we have to establish efficient animal models to test the efficacy of novel therapeutic strategies and anticipate and overcome resistance mechanisms.

Our in vivo PDAC CRISPR/Cas9-screen identified several bona fide PDAC tumor suppressor genes, such as *USP15* and *SCAF1*. USP15 is a broadly expressed deubiquitinase and was implicated in several cancer-associated pathways. For example, USP15 can act as a tumor promoter in estrogen receptor-positive breast cancer by deubiquitinating and thereby stabilizing the estrogen receptor[51], by stabilizing TGF-β receptor 1 (TGFβR1) in glioblastoma[31], or by deubiquitinating and stabilizing MDM2, leading to p53 inactivation[28]. USP15 also plays important roles in regulating TNFα-signaling by inhibiting the proteolysis of TAK1-TAB2/3 complex[35] or regulating IκBα[32]. In addition, USP15 also controls inflammation in response to infectious and autoimmune insults and following tissue damage[52]. In line with these reports, we found that loss of USP15 in pancreatic epithelium leads to reduced TGF-β signaling and downregulation of inflammatory responses to cytokine and chemokines such as TNFα and IL6 signaling.

In pancreas cancer cell lines, Peng et al. showed that USP15 regulates homologous recombination and DNA double-strand break (DSB) repair by deubiquitinating BARD1, thereby promoting BARD1-

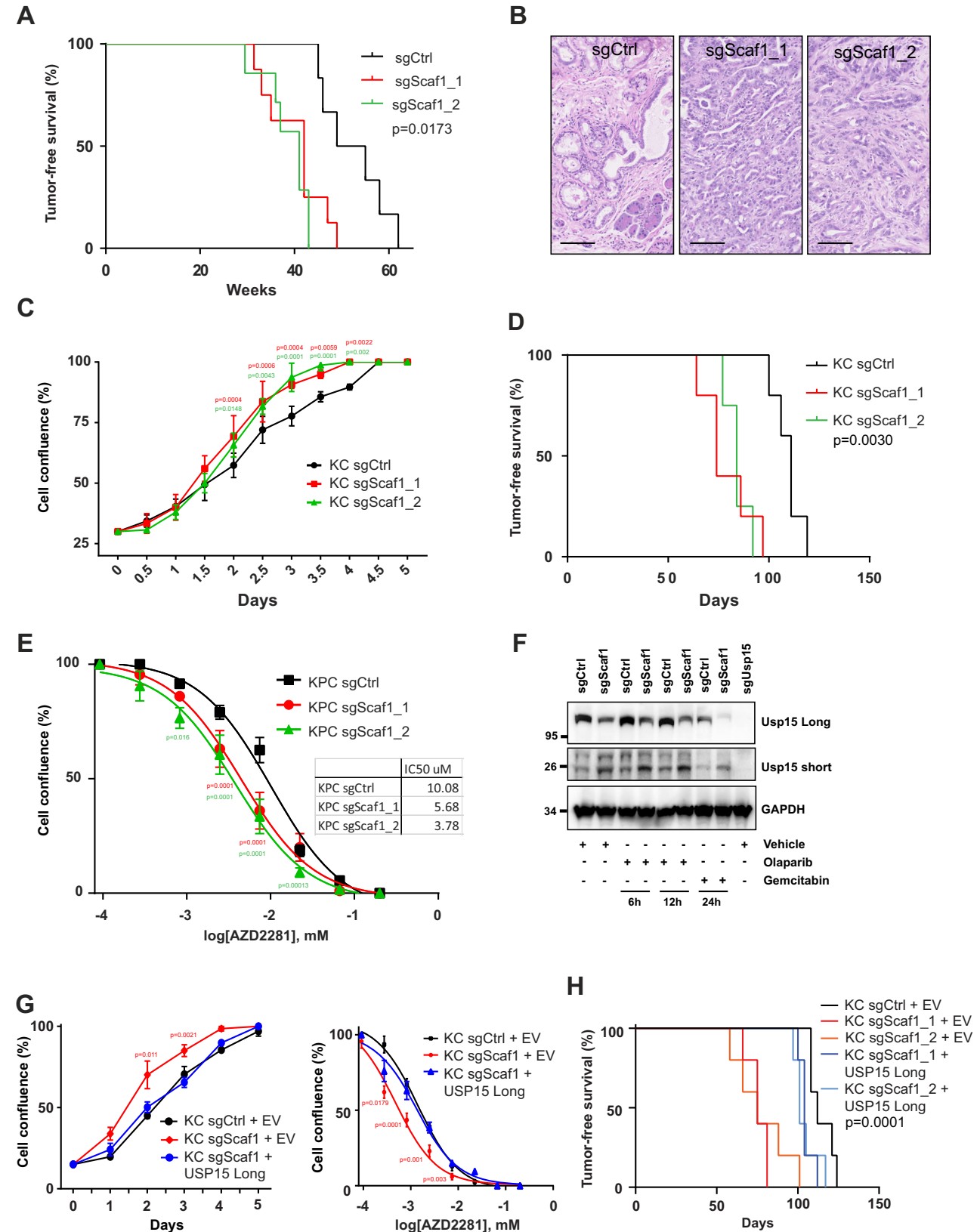

HP1γ interaction and increased BARD1-BRCA1 retention at DSB. Mutation or loss of USP15 impairs DSB repair and thus leads to increased sensitivity to PARP inhibition[36]. We recapitulated these data and also showed increased sensitivity to PARP inhibition but also increased sensitivity to gemcitabine, a common PDAC chemotherapy. The increased sensitivity to Gemcitabine was surprising at first, as

Gemcitabine does not induce DNA DSB. Transcriptional profiling of *USP15* knock-out cells showed that glutathione metabolism, oxidative stress, and redox pathways are significantly upregulated, indicating that *USP15* knock-out cells are experiencing increased cellular stress. This could conceivably further explain the increased sensitivity to Olaparib but also to Gemcitabine.

**Fig. 4 | Scaf1 functions as PDAC tumor suppressor. A** Tumor-free survival of Pdx1-Cre;LSL-*Kras*^G12D;LSL-*Cas9-GFP* mice injected with CRISPR AAV targeting the indicated gene or non-targeting control sgRNA (sgCtrl $n = 6$). Two independent sgRNAs were used (sgScaf1_1 $n = 8$, sgScaf1_2 $n = 7$). Log-Rank test (Mantel-Cox). **B** Representative H&E images showing multifocal PanINs in sgCtrl-transduced pancreas and PADC tumors in sgScaf1-transduced pancreas. Scale bar 100 μm. **C** Cell proliferation curves of KC sgCtrl and sgScaf1 cells were obtained using the IncuCyte live-cell imaging and data are expressed as cell confluence percentage (%; mean ± SD, $n = 3$ independent experiments). Two-way ANOVA (sgScaf1_1 $p = 0.0039$: sgScaf1_2 $p = 0.00042$), Dunnett's multiple comparison. **D** Tumor-free survival of NSG mice orthotopically injected with sgCtrl ($n = 5$) or sgScaf1_1/2 ($n = 5$; $n = 5$) KC cells. Two independent sgRNAs were used. Log-Rank test (Mantel-Cox). **E** Dose-response curves for KPC sgCtrl or sgScaf1 cells treated with the indicated concentration of Olaparib (mean ± SD, $n = 3$ independent experiments). Two-way ANOVA (sgScaf1_1 $p = 0.0084$: sgScaf1_2 $p = 0.0028$), Dunnett's multiple comparisons. **F** Representative Western Blot of Usp15 in KC cells transduced with the indicated sgRNAs and treated as indicated. This experiment was repeated independently two times with similar results. **G** Cell growth curves of KC sgCtrl and sgScaf1 cells expressing the listed isoform of USP15 or an empty vector (EV). Data are expressed as cell confluence percentage (%; mean ± SD, $n = 3$ independent experiments); two-way ANOVA (sgScaf1+EV p = 0.0342), Dunnett's multiple comparison. Dose-response curves for KC sgCtrl and sgScaf1 cells expressing the listed isoforms of USP15 or EV and treated with Olaparib. (%; mean ± SD, $n = 3$ independent experiments; two-way ANOVA (sgScaf1+EV $p = 0.00129$), Sidak multiple comparison **H** Tumor-free survival of NSG mice orthotopically injected with sgCtrl ($n = 5$) or sgScaf1 KC cells expressing the listed isoforms of USP15 ($n = 5$: $n = 5$) or EV ($n = 5$; $n = 5$). Log-Rank test (Mantel-Cox).

In addition to USP15's role in regulating sensitivity to PARP inhibition, we found that USP15 functions as a strong tumor suppressor in pancreatic cancer. Importantly, our data indicates that USP15 functions in a haploinsufficient manner, which is seen in a clinically relevant portion of ~25% of PDAC patients. The growing list of haploinsufficient cancer driver genes identified in genetic in vivo screens[11,53–58] raises the provocative question of whether the lack of comprehensive screening within innate tumor microenvironment obstructed our capabilities of identifying many of these haploinsufficient cancer driver genes. This is in line with recent findings from Martin et al., showing that the adaptive immune system is a major driver of selection for tumor suppressor gene inactivation[59]. Historically, most attention has focused on frequently mutated dominant oncogenes and recessive tumor suppressor genes, but recent large-scale genomic efforts revealed recurrent copy number alterations (CNA) mainly involving shallow losses or gains of large regions[60,61]. The remarkably recurrent, specific pattern of these CNAs certainly indicates that one or several genes in these regions are being selected for, presumably by loss of haploinsufficient tumor suppressor genes[62]. However, with a few exceptions[63–66], cancer driver genes conferring selective advantage of certain CNA are virtually unknown. Bioinformatic approaches to delineate cancer driver from passenger mutations are usually based on statistical enrichment of specific patterns of somatic point mutations and/or amino acid conservation signifying functional importance. However, CNA are simply too large sometimes spanning hundreds to thousands of genes, too numerous and too noisy and most studies are underpowered to call driver genes by bioinformatic means. Given that some CNAs are linked to worse outcomes and might have therapeutic implications, functional annotation of recurrent haploinsufficient cancer driver genes in recurrent CNAs is of high clinical relevance. For example, it will be interesting to see whether USP15 functions as a haploinsufficient tumor suppressor in other cancers that show frequent shallow USP15 deletion such as sarcoma, esophageal adenocarcinoma, melanoma or lung cancers Supplementary Data Fig. 9a).

The second new PDAC tumor suppressor gene identified in this study is SCAF1. SCAF proteins are part of a superfamily of pre-mRNA splicing factors. Surprisingly, SCAF proteins were recently shown to be essential to regulate alternative polyadenylation. Gregersen et al. recently showed that SCAF1, 4, 8 and 11 are part of the elongating RNA polymerase II complex and that SCAF4 and SCAF8 coordinate the transition between elongation and termination, ensuring correct polyA site selection and RNAPII transcriptional termination. Loss of SCAF8 and especially loss of SCAF4 and SCAF8 leads to pre-mature polyadenylation and the production of shorter protein isoforms[43]. Interestingly, Gregersen et al. showed that the USP15 gene is subject to alternative polyadenylation and that SCAF8- and SCAF4/8-knockout cells exhibit a short ~30 kDa USP15 isoform at the expense of the longer isoform. Given that SCAF1 is a close homolog of SCAF4/8 and also part of the RNA polymerase II complex, it is interesting to speculate whether the shorter ~30 kDa USP15 isoform with concomitant reduction of the long USP15 isoform observed in Scaf1 knock-out pancreatic cancer cells might also be the result of premature polyadenylation. Interestingly, expression of the long USP15 isoform rescued the SCAF1 knock-out phenotypes, providing a direct molecular link between our top two new hits. Further functional studies will be required to explore the relevance of SCAF1 and aberrant polyadenylation in USP15 processing and pancreatic cancer.

Finally, it would be of great significance to test whether PDAC patients with *USP15* or *SCAF1* alterations indeed show increased sensitivity and a better therapeutic outcome to Gemcitabine and PARP inhibition, as indicated by our findings. Interestingly, PARP inhibition induced decreased hedgehog signaling in USP15 and SCAF1 knock-out cells and given that preclinical studies indicated that hedgehog pathway inhibition reduces growth and metastasis of pancreatic ductal adenocarcinoma cells[67–69], it will be interesting to explore whether PARP inhibition would synergize with hedgehog inhibition in USP15 or SCAF1 loss-of-function setting. Together, this study highlights the utility of in vivo CRISPR screening to integrate cancer genomics and mouse modeling for rapid discovery, validation and characterization of PDAC genes and vulnerabilities.

## Methods
### Animals
Animal husbandry, ethical handling of mice and all animal work were carried out according to guidelines approved by the Canadian Council on Animal Care and under protocols approved by the Centre for Phenogenomics Animal Care Committee (18-0272H). The animals used in this study were Pdx1-Cre [B6.FVB-Tg(Pdx1-cre)6Tuv/J] in a FVBN background and KrasG12D/+ mice [B6.129-Krastm4Tyj] in a mixed C57/Bl6 background and R26-LSL-Cas9-GFP [#026175] in a C57/Bl6 background, all obtained from Jackson laboratories. Conditional USP15-tm1c mice [C57BL / 6N-Usp15tm1c(EUCOMM)Wtsi / Tcp] were kindly provided by Philippe Gros. CRISPR screens in the Pdx1-Cre;KrasG12D/+; Cas9 cohort were performed in a mixed FVBN/C57Bl6 background and equal number of female and male mice were used. Genotyping was performed by PCR using genomic DNA prepared from mouse ear punches. When total tumor mass per animal exceeded 1000mm3, mice were monitored bi-weekly and scored in accordance with SOP "#AH009 Cancer Endpoints and Tumour Burden Scoring Guidelines". Maximal tumor size permitted by the animal ethics committee and according to SOP AH009 must not exceed 1700mm3, which is approximately a 1.5 cm×1.5 cm tumour and which was never exceeded in the current study.

### Adeno-associated virus constructs and library construction
sgRNAs targeting pancreatic cancer long tail genes were obtained from Hart et al.[70], (4 sgRNAs/gene) and non-targeting sgRNAs were obtained from Sanjana et al.[71], ordered as a pooled oligo chip (CustomArray Inc., USA) and cloned into AAV sgRNA-H2B-RFP engineered from AAV:ITR-

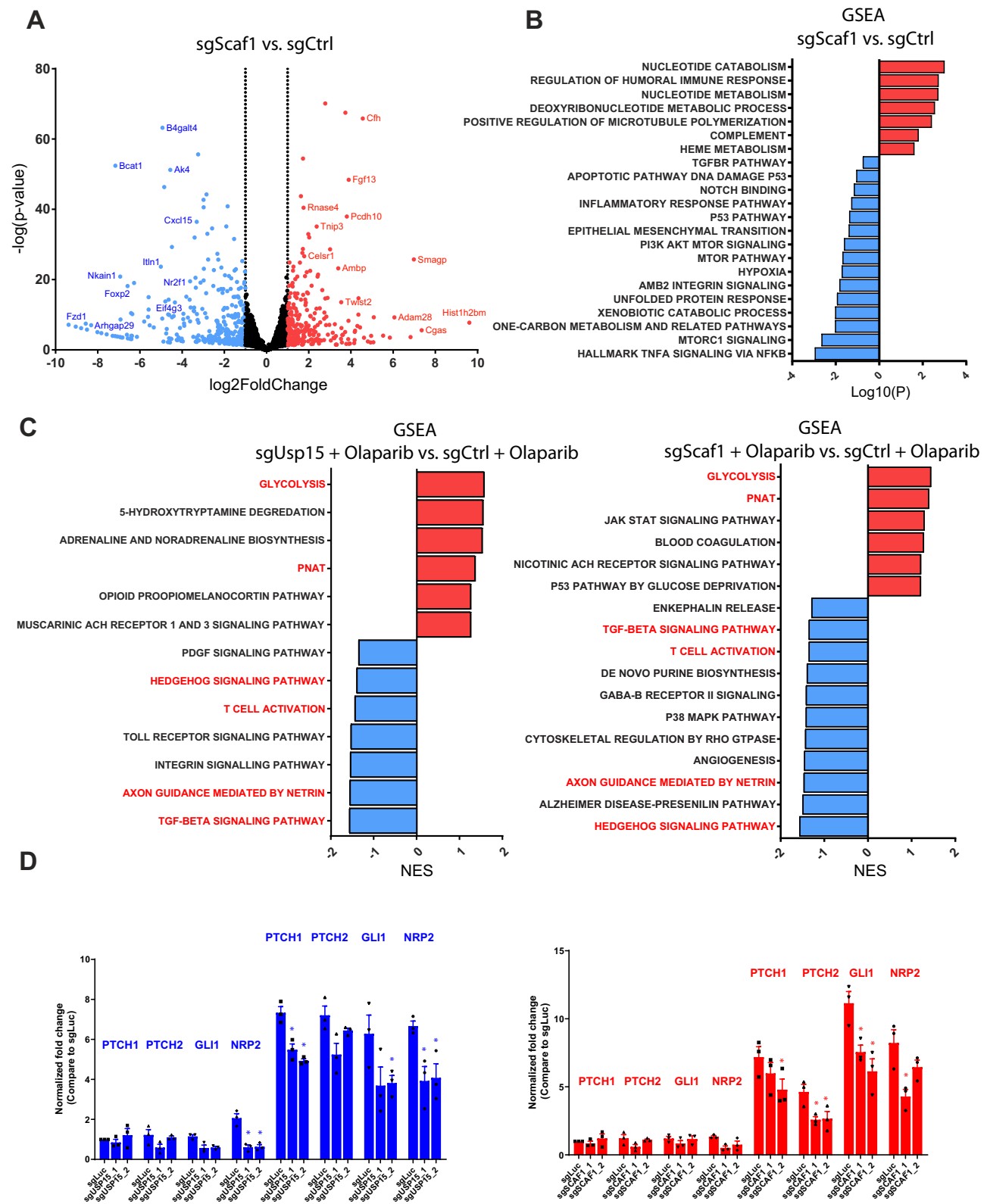

**Fig. 5 | Scaf1 regulates TNFa and p53 signaling as well as hedgehog signaling in response to Olaparib. A** Volcano Blot showing differential expressed genes between Scaf1-knockout compared to control KC cells. Wald test and Benjamini-Hochberg (BH)-adjusted P-value. Two independent sgRNAs, two biological duplicates. **B** Bar graph showing gene set enrichment analysis of Scaf1-knockout compared to control KC cells. GSEA nominal p-values. Two independent sgRNAs, two biological duplicates. **C** Bar graph showing gene set enrichment analysis of Usp15-knockout and Scaf1-knockout compared to sgCtrl control KC cells treated with Olaparib (1 μM). GSEA nominal p-values. Two independent sgRNAs, two biological duplicates. **D** Expression levels of genes related to HH signaling evaluated by RT-qPCR in the indicated KC cell lines. Results were normalized with Gapdh and are expressed in fold change to CTRL (mean ± SEM, n = 3 independent experiments). Cells were treated with 100 nM Smoothened Agonist (SAG) and 1 μM Olaparib. Two-sided T-test, for sgUSP15: NRP2 $p = 0.021/p = 0.018$; PTCH1 $p = 0.037/p = 0.033$; GLI1 $p = 0.033$; NRP2 $p = 0.042/p = 0.037$; for sgScaf1: PTCH1 $p = 0.046$; PTCH2 $p = 0.040/p = 0.042/p = 0.033$; GLI1 $p = 0.037/p = 0.031$; NRP2 $p = 0.042$.

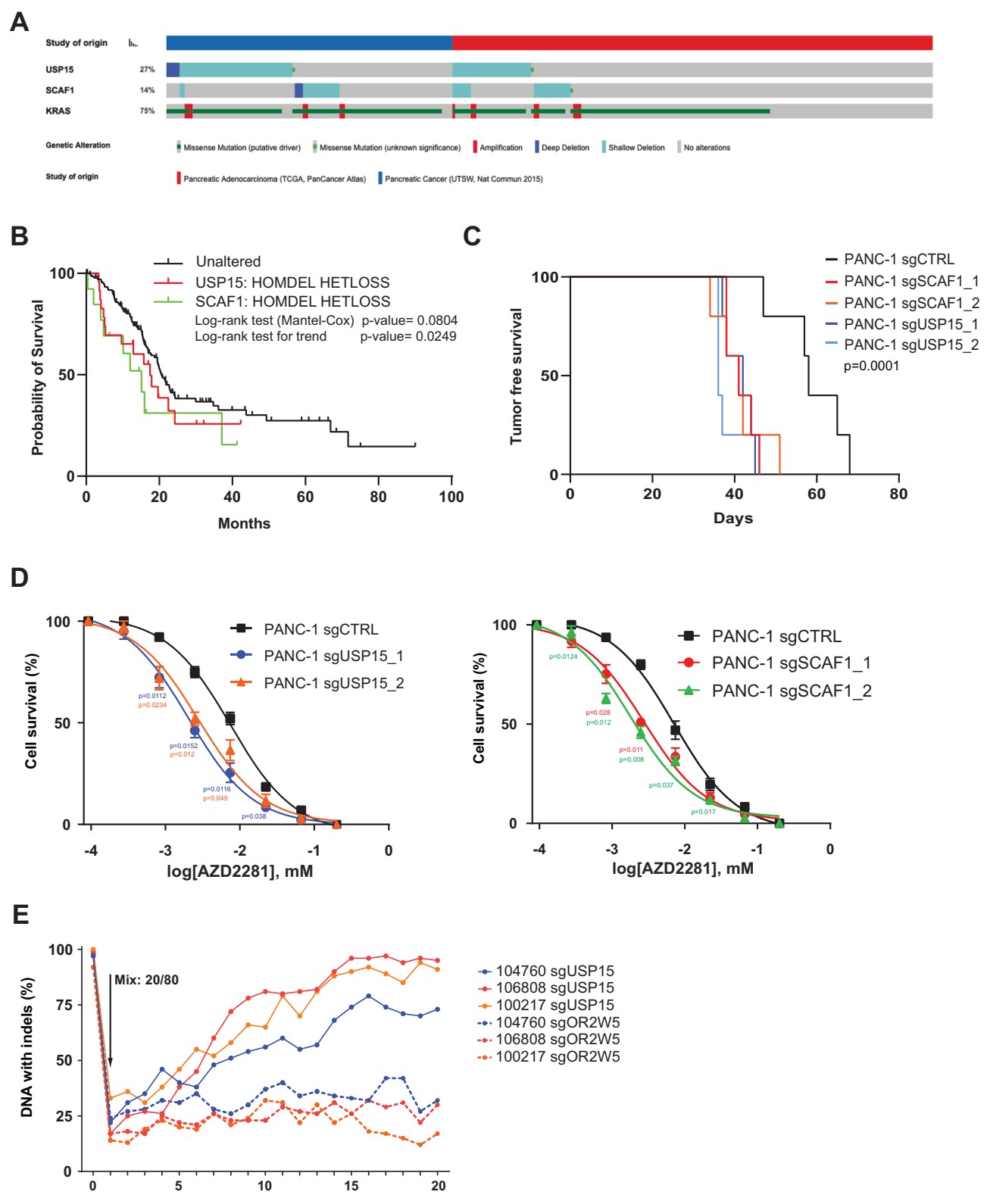

**Fig. 6 | USP15 and SCAF1 function as tumor suppressor in human pancreatic cancer. A** Oncoprint of the indicated genes in PDAC samples (*n* = 293, TCGA). **B** Kaplan-Meier survival analyses of PDAC patients with deep or shallow *USP15* or *SCAF1* deletion. (*n* = 293, TCGA) Log-Rank test (Mantel-Cox). **C** Tumor-free survival of NSG mice orthotopically injected with sgCtrl (*n* = 5), sgUsp15_1/2 (*n* = 5; *n* = 5) or sgSCAF1_1/2 (*n* = 5; *n* = 5) PANC-1 cells. Two independent sgRNAs were used. Log-Rank test (Mantel-Cox) **D** Dose-response curves for sgCtrl, sgUsp15 or sgSCAF1 PANC-1 cells treated with Olaparib. (%; mean ± SD, *n* = 3 independent experiments) two-way ANOVA (sgUsp15_1 *p* = 0.0242; sgUsp15_2 *p* = 0.0387; sgScaf1_1 *p* = 0.0281; sgScaf1_2 *p* = 0.0371), Dunnett's multiple comparison. **E** sgUSP15 and sgOR2W5 PDO Competition Assay. sgUSP15 and control sgOR2W5 patient-derived organoids were disassociated into single cells and mixed in a 20:80% ratio. Organoid cultures were passaged, and a sample was collected every ~7 days. Percentage of DNA indels is tracked over time by sanger-sequencing and TIDE analysis.

U6-sgRNA(backbone)-pCBh-Cre-WPRE-hGHpA-ITR kindly provided by Feng Zhang (Addgene plasmid #60229) We excluded frequent and known pancreatic cancer tumor suppressor genes such as *TP53* or *Smad4* from the Cancer long tail genes library. The non-targeting sgRNAs are those designed not to target the mouse genome as a negative control. Ad-Cre and Ad-GFP were purchased from the Vector Core at the University of Iowa.

## AAV production and transduction

293AAV cells (AAV-100, Cell Biolabs inc) were seeded on a poly-L-lysine coated 15 cm plates and transfected using PEI (polyethyleneimine) method in a non-serum media with AAV construct of interest along with AAV packaging plasmids pAAV-DJ Vector and pHelper Vector. 8 hours post-transfection media was added to the plates supplemented with 10% Fetal bovine serum and 1% Pencillin-Streptomycin antibiotic solution (w/ v). 48 hours later, the viral supernatant and cell pellet were collected. Cell lysis was performed by four rounds of freeze/thaw cycles using a dry ice/ethanol bath and filtered through a Stericup-HV PVDF 0.45-µm filter, and then concentrated ~2,000-fold by ultracentrifugation in a MLS-50 rotor (Beckman Coulter). Viral titers were determined by infecting the R26-LSL-tdTomato MEFs and FACS-based quantification. In vivo viral transduction efficiency was determined by injecting decreasing amounts of a single viral aliquot of known titer, diluted to a constant volume of 10 µl per pancreas. We collected pancreas at 7 days post-infection and determined percent infection using FACS.

## Pancreas viral transduction

Four to 6-week-old mice were anesthetized using 3% isoflurane. Mice were subjected to laparotomy and injected with 10 uL of purified AAV solution resuspended in PBS using a 28-gauge needle through the head and tail of the pancreas by slowly retracting the needle. Successful administration was confirmed by a uniform swelling of the injected area. Laparotomies were subsequently closed with a two-layer suture. The majority of the mice survived this operation with no observed complications. Mice were then sacrificed at various days post-injection, and pancreatic tissue was harvested for DNA extraction and immunohistochemistry. To verify the sgRNA abundance and representation in the control and pancreas long-tail genes libraries, MEFs were transduced with library virus and collected 48 h post-transfection. Genomic DNA from all samples was extracted using a QIAamp DNA tissue mini kit (Qiagen). Barcode pre-amplification, sequencing and data processing were performed as described below. Ad-Cre and Ad-GFP were injected at a final pfu of 1.10^7 pfu/mL.

## Deep Sequencing: sample preparation, pre-amplification and sequence processing

Genomic DNA from epithelial and tumor cells was isolated with the DNeasy Blood & Tissue Kit (Qiagen). 5 µg genomic DNA of each tumor was used as a template in a pre-amplification reaction using a unique barcoded primer combination for each tumor with 20 cycles and Q5 High-Fidelity DNA Polymerase (NEB). The following primers were used:

FW:5′AATGATACGGCGACCACCGAGATCTACAC**TATAGCCT**A-CACTCTTTCCCTACACGACGCTCTTCCGATCTTGTGGAAAGGACGAAACACCG-3′

RV:5′CAAGCAGAAGACGGCATACGAGAT**CGAGTAAT**GTGACTGGAGTTCAGACGTGTGCTCTTCCGATCTATTTTAACTTGCTATTTCTAGCTCTAAAAC-3′

The underlined bases indicate the Illumina (D501-510 and D701-712) barcode location that were used for multiplexing. PCR products were run on a 2% agarose gel, and a clean ~200 bp band was isolated using Zymo Gel DNA Recovery Kit as per manufacturer instructions (Zymoresearch Inc.). Final samples were quantitated and then sent for Illumina Next-seq sequencing (1 million reads per tumor) to the sequencing facility at Lunenfeld-Tanenbaum Research Institute (LTRI). Sequenced reads were aligned to sgRNA library using Bowtie version

1.2.2 with options −v 2 and −m 1. sgRNA counts were obtained using MAGeCK count command. A detailed cloning protocol can be found in Loganathan et al.[72].

## Analysis of genome editing efficiency

LSL-Cas9-GFP MEFs, KPC-LSL-Cas9-GFP and KC-LSL-Cas9-GFP were cultured and infected with AAV carrying corresponding sgRNAs. Cells were live sorted for GFP + /RFP+ expression and expanded further to extract genomic DNA using DNeasy Blood & Tissue Kit (Qiagen). Genomic DNA from tumors from the mice injected with single sgRNAs was also isolated using the same kit. PCR was performed by flanking the regions of sgRNA on genomic DNA from both WT cells and cells infected with respective viruses or tumors and sent for Sanger sequencing. Gene editing efficiency was determined by Tracking of Indels by Decomposition (TIDE https://tide.nki.nl) algorithm.

## qRT-PCR

RNA samples were purified from cells using either Trizol (Life Technologies), treated with ezDNase (Invitrogen), and reverse transcribed into cDNA using SuperScript IV VILO (Invitrogen). Primers were designed to span exon junctions using Primer3Plus. Primers used are shown in Supplementary Data 5. Primers were validated against a standard curve and relative mRNA expression levels were calculated using the comparative Ct method normalized to the housekeeping mRNA (Gapdh and Ppib). Real-time quantitative PCR (qRT-PCR) reactions were performed in 384-well plates containing 12.5 ng cDNA, 150 nM of each primer, and 5 µl of SYBR Green in a 10 uL total volume reaction. PowerUP SYBR Green Master Mix (Applied Biosystems) was used with the QuantStudio 5 (Applied Biosystems). The cycling time used was as per manufacturing protocol, and each reaction was performed in technical triplicates.

## Western blot analysis

Cells were lysed using RIPA buffer (50 mM Tris HCl, pH 8.0, 150 mM NaCl, 1 mM EDTA, 1% NP-40 and 0.25% deoxycholate) supplemented with protease inhibitor tablets (Roche Molecular Biochemicals) and phosphatase inhibitors (1 mM NaF, 1 mM Na3VO4 and 1 mM b-glycerophosphate). Fifty µg of proteins were separated by sodium dodecyl sulfate-polyacrylamide gel electrophoresis (SDS-PAGE) and probed overnight, 4°C with the stated antibodies, then visualized by electrochemiluminescence (ECL, Roche Molecular Biochemicals).

## Antibodies

The following primary antibodies were used in this study: Anti-Cytokeratin 19 antibody [RCK108] (1:200, Abcam ab9221), Anti-USP15 monoclonal antibody (M01), clone 1C10 (1:500, Novus Biological H00009958-M01), Anti-GAPDH (6C5) (1:1000, Santa Cruz sc-32233), p21 (1:200, Santa Cruz sc-6246), P53 (1:500, Santa Cruz (DO-1): sc-126), MDM2 (1:200, Santa Cruz sc-965), NRF2 (1:500, Cell signaling D1Z9C).

## Wound Healing Assay

KC cells were seeded in 12-well plates. After overnight culture, the culture medium was changed to DMEM containing 0.1% FCS. Wounds were made by scraping a plastic pipette tip across the cell monolayer, and wounded cells were cultured with 10 ng/ml TGF-b1 for 48 h. Phase contrast images were recorded at the time of wounding (0 h), 24 h, and 48 h. Wound areas were quantified using ImageJ. Wound healing was estimated as a percentage of the remaining wound area relative to the initial wound area.

## IC50 determination and cell viability assay

Cells were seeded in 24-well plates. Drugs were added the next day in triplicates. Following 5 days of drug treatment, cell viability was assessed using PrestoBlue™ Cell Viability Reagent according to the manufacturer's instructions. Data analysis and calculation of the half-

maximal inhibitory concentration (IC50) were carried out using GraphPad Prism (GraphPad).

## Cell culture

Primary mouse tumor cells KPC and KC were cultured in DMEM supplemented, 10% FBS and Pen Strep. Panc1 cells were cultured in DMEM supplemented, 10% FBS and Pen Strep. Cells were cultured in monolayer for growth and transfection with AAV CRISPR construct containing Cre or H2B-RFP resistance and sgRNA targeting genes of interest. Cells were tested for cutting efficiency post-selection with TIDE described earlier and by western blot. Panc1 and MiaPaca2 were obtained from ATCC.

## Immunofluorescence

Tissue sections were fixed with 4% paraformaldehyde for 10 minutes. Following fixation, slides were rinsed 3 times with PBS for 5 minutes. For cells, permeabilization was carried out using 0.5% Tween-20 in PBS at 4 °C for 20 minutes and rinsed with 0.05% Tween-20 in PBS for 5 minutes, 3 times each at room temperature. Samples were blocked at room temperature with blocking serum (recipe: 1% BSA, 1% gelatin, 0.25% goat serum 0.25% donkey serum, 0.3% Triton-X 100 in PBS) for 1 hour. Samples were incubated with primary antibody diluted in blocking serum overnight at 4 °C followed by 3 washes for 5 minutes in PBS. The secondary antibody was diluted in blocking serum with DAPI and incubated for 1 hour at room temperature in the dark. Following incubation, samples were washed 3 times for 5 minutes in PBS. Coverslips were added on slides using MOWIOL/DABCO-based mounting medium and imaged under the microscope the next day. For quantification, laser power and gain for each channel and antibody combination were set using secondary-only control and confirmation with primary positive control and applied to all images.

## PDO CRISPR Knockout and Competition Assay

Patient-derived organoid (PDO) lines with varying USP15 expression (medium, and high) were chosen. Lines were established using patient samples with confirmed PDAC diagnosis acquired from surgical specimens or image-guided percutaneous core needle biopsies. Tumour tissue from biopsy and resection was enzymatically and mechanically digested and mixed in a gel-like matrix known as Matrigel and allowed to grow as organoids following a modified protocol from Boj et al. [73] (digestion overnight at 4 °C instead of at 37 °C). Three selected PDO lines were first disassociated from organoids into single cells by adding TrypLE (GIBCO) until completely disassociated. Cells are pelleted by spinning at 300 g x 10 min at 4 °C. The pellets are aspirated and resuspended in a minimal volume of Advanced DMEM/F-12 (GIBCO). The cells are then counted, and the desired amounts of cells are aliquoted for CRISPR ribonucleoprotein (RNP) electroporation (~50 K – 300 K cells per condition). For these experiments, each sample had 3 different CRISPR conditions. These conditions included CRISPR targeting USP15, a control gene OR2W5, and unedited (UE) control cells that did not undergo CRISPR electroporation. CRISPR electroporation was done using chemically synthesized crRNAs (IDT), Alt-R® CRISPR-Cas9 (IDT), and the 4D-Nucleofector (Lonza). After electroporation, 180 uL prewarmed organoid media was added and each condition was transferred to a 1.5 mL tube spun down at 300 g x 10 min at 4 °C and resuspended in Matrigel. Cells were then plated in 50ul Matrigel domes (1 dome per well) in a 24-well tissue culture plate (Nunc), left to set at 37 °C for 15 mins then 500 ul of organoid media was added to each well. To determine if USP15 expression provided a tumour growth or proliferation advantage a competition assay was conducted for three PDO samples. 72 hours after CRISPR, organoids are disassociated into single cells (as described above) and mixed with UE control in a 20/80% ratio. Cell collection occurred every ~7 days. Cells collected for passaging are spun down and resuspended in Matrigel as described

above, depending on confluency 1-4 domes are plated. Informed consent was obtained from all the patients to generate PDOs at the University Health Network, Canada.

## RNA-seq and GSEA analyses

RNA was extracted from cells using Quick-RNA plus mini Kit (Zymoresearch Inc., #R1057) as per the manufacturer's instructions. RNA quality was assessed using an Agilent 2100 Bioanalyzer, with all samples passing the quality threshold of RNA integrity number (RIN) score of >7.5. The library was prepared using an Illumina TrueSeq mRNA sample preparation kit at the LTRI sequencing Facility, and complementary DNA was sequenced on an Illumina Nextseq platform. Sequencing reads were aligned to mouse genome (mm10) using Hisat2 version 2.1.0 and counts were obtained using featureCounts (Subread package version 1.6.3)[74]. Differential expression was performed using DESeq2[75] release 3.8. Gene set enrichment analysis was performed using GSEA version 4.0; utilizing genesets obtained from MSigDB (https://www.gsea-msigdb.org/gsea/msigdb) with a NES cutoff of +/- 1.4. For integration with human and existing mouse tumor models, clustering was conducted after normalization and filtering for only intrinsic genes as described previously[75].

## Statistics and reproducibility

All quantitative data are expressed as the mean ± SEM. Differences between groups were calculated by two-tailed Student's t-test, Wilcoxon Rank-Sum test (when data was not normally distributed) or Log-rank test for survival data using Prism 7 (GraphPad software).

## Reporting summary

Further information on research design is available in the Nature Portfolio Reporting Summary linked to this article.

# Data availability

The RNA-seq data generated in this study have been deposited in the NCBI Gene Expression Omnibus under GEO accession GSE220556. All remaining data can be found in the Article, Supplementary and Source Data files.

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

## Acknowledgements

We thank all members of our laboratories for their helpful comments, with additional thanks to Y.Q. Lu, and G. Mbamalu for their insight and assistance. We also thank The Centre for Phenogenomics, Network Biology Collaborative Centre and Flow Cytometry facility at LTRI as well as the Flow Cytometry Facility at the University of Toronto. The UbV variants were kindly provided by Sidhu S. Sachdev. Funding: This study was conducted with the support of the Ontario Institute for Cancer Research (PanCuRx Translational Research Initiative) through funding provided by the Government of Ontario (D.S., F.N., S.G.) the Wallace McCain Centre for Pancreatic Cancer supported by the Princess Margaret Cancer Foundation (S.G.), the Terry Fox Research Institute (S.G.), the Canadian Cancer Society Research Institute (S.G.), Pancreatic Cancer Canada (S.G.) and a Canadian Institute of Health project grant to K.C. and D.S. (PJT175270). This research was undertaken, in part, thanks to funding from the Canada Research Chairs Program to D.S. The development of *Usp15*$^{flox/flox}$ mice was supported in part by a consortium grant from the Healthy Brains for Healthy Lives McGill program, the Consortium Quebecois de la Recherche sur le Médicament, Brain Canada, and Corbin Therapeutics. Selected artwork (syringe, mouse, pancreas, human shape, DNA helix and tumors schematics shown in the figures were used from or adapted from pictures provided by Servier Medical Art (Servier; https://smart.servier.com/), licensed under a Creative Commons Attribution 4.0 Unported License.

## Author contributions

S.M. performed all experiments. M.G., A.M., and K.C. performed bioinformatics analysis of the RNAseq experiments. T.W and F.N performed the PDO experiments. AZ, GHJ and F.N. analysed the human PDAC patient samples. D.D., K.N.A. and R.T. helped with mouse and RT-PCR experiments. N.F. and P.G provided the Usp15$^{fl/fl}$ mice and experimental guidance and helped with experimental design. R.W., A.S., G.G.N., S.G. and U.E. provided guidance on project and experimental design. D.S. coordinated the project and the experiments and together with S.M. wrote the manuscript.

## Competing interests

The authors declare no competing interests.
