## [Peer Review File · Nature Communications]

Reviewers' Comments:

Reviewer #1:

Remarks to the Author:

In this manuscript, Martinez et al. conducted in vivo CRISPR/Cas9 screens in order to identify novel pancreatic tumor suppressors. Within a library targeting the recurrently mutated pancreatic cancer genes, authors identified USP15 and SCAF1 as tumor suppressor genes. The authors then validated the tumor suppressor function of USP15 in vitro and in vivo conditions and showed that USP15 regulates several immune-related pathways involved in PDAC development. They also provided evidence that the loss of SCAF1 results in the formation of a truncated USP15 isoform. The authors also showed that loss of USP15 or SCAF1 sensitizes cells to PARP inhibition and Gemcitabine.

Overall, the manuscript recommended a mechanism important for PDAC development and therapy response. One of the study's major strengths is conducting the screenings in vivo, which allowed the authors to identify targets under physiologically relevant conditions. The manuscript is well-written except for a few minor errors stated below. There are a few points that could further enhance conclusions and improve the novelty of the findings in this manuscript forward.

- Authors showed the role of USP15 and SCAF1 alterations, however, they did not provide the actual patient data to support their findings. By analyzing the PDAC cohorts and stratifying the patients based on their USP15 and SCAF1 mutation status (loss of function or gain of function), they can show if there is a correlation with the survival. Since it has been stated that these alterations are observed in 31% of the patients, they would potentially have enough sample size for this analysis.
- Authors defined the "long-tail" as recurrent but less frequent alterations. They wanted to identify the tumor suppression relevant roles of this long-tail PDAC genes. However, alteration of the genes identified is represented in 31% of the patient population. Since this is a considerable percentage, authors should more clearly explain the cut-off for mutation frequency.
- Authors showed that loss of SCAF1 results in the formation of a truncated USP15 isoform and provided rescue experiments where they combined sgScaf1 with USP15 long isoform in vitro. Since this is one of the most important findings of the manuscript, and the hit genes are identified from an in vivo screening set-up, authors should also provide in vivo evidence and conduct similar rescue experiments in vivo.
- The connection between the depletion of SCAF1 and the depletion of the long form of USP15 is interesting and warrants further investigation. At a minimum, the connection should be further delineated with rescue experiments. For example, will overexpression of USP15 in SCAF1-depleted cells rescue the SCAF1 depletion phenotype?
- Are these results (the connection between SCAF1 and USP15) supported by the patient data? If they sorted TCGA PDAC patients into SCAF1 high and low, will this be correlated with the USP15 long and short isoforms?
- If depletion of SCAF1 leads to deletion of USP15 long isoform, one might expect a mutual exclusivity in terms of loss of function mutations between these two genes. Is this true in human PDAC data?
- Authors showed that loss of USP15 or SCAF1 sensitizes cells to PARP inhibition and Gemcitabine in vitro; however, showing that this sensitization works in tumor models would be an important experiment as well.
- In Supplementary Figure 5e, authors showed the presence of a short USP15 isoform in human Panc1 cells and provided this as evidence for this isoform to be evolutionary conserved. Authors should validate that this isoform is present in a panel of human pancreatic cancer cell lines to support this data.
- How does the sRNA detection frequency of each gene in vivo correlate with the mutation

frequency of the target gene in TCGA data? Can it be concluded that the most frequent genes are also the ones that can form tumors most frequently?

- Finally, the Authors never talked about the Mets in liver and lungs. The model systems look like a great platform to discover if certain sgRNAs are selectively enriched in specific metastasis locations. Authors should analyze metastatic lesions and present findings about the metastatic potential of sgRNAs targeting various genes.

Minor points:

- In the text body, Figure 2e is labeled as Figure 2c, and Supplementary Figure 3f is labeled as Supplementary Figure 3e.

Reviewer #2:

Remarks to the Author:

Lines 62-63: "USP15 and SCAF1 alterations are observed in 31% of pancreatic cancer patients". Is this true? Please provide a reference.

Line 165: "further supporting their functions as strong suppressors of pancreatic cancer". This is a strong statement based on one mouse model (KC). Perhaps rephrase to "further supporting their functions as strong suppressors of pancreatic cancer in KC mice"?

217-219: "analyses (GSEA) revealed significantly upregulated gene sets associated with xenobiotic detoxification, glutathione metabolism, anabolic processes, and oxidative phosphorylation". What is known about these pathways in pancreatic cancer (line 230)?

Line 514: "RNA-seq and GSEA analyses". It is not clear how many replicates were used in the RNA-seq analysis and whether the replicates were biological or technical. Based on Fig. 3, I assume that one biological replicate was used for each sgRNA. For differential gene expression analysis, two samples are not enough. Also, I would think at least two technical replicates are needed for each sgRNA. It would also be helpful to show the concordance in global gene expression between KC cell lines targeted by the two sgRNAs.

Line 216, "(FDR)<0.05", please specify the FDR method used.

Line 219-220, "These findings are in line with USP15's known role in negatively regulating NRF2 (encoded by the NFE2L2 gene)", please provide a reference.

Line 224, "GSEA also revealed decreased gene sets". Do you mean "depleted gene sets"? Were those also up-regulated genes?

Line 514: please specify the cutoff used for determining enriched gene sets (NES cutoff)

Figure 3C, I am puzzled by the heatmap bars (NES). Enrichment scores only apply to gene sets, not individual genes. What is shown in the heatmaps – gene expression levels or normalized enrichment score (NES)?

Also, in Figure 3C, the authors showed the enrichment plots for two pathways. The most significant pathway with deleted gene set – the cytokine receptor binding – is not shown. Conversely, none of significant pathways with enriched gene sets were shown. I am curious about how these choices were made.

Figure 5C, genes in glycolysis pathway were significantly enriched in both sgUsp15 vs sgCtrl+Olaparib and sgScaf1 vs sgCtrl+Olaparib comparisons. The authors highlighted pathways with enriched/depleted genesets common to both comparisons. Since little in common was found between the comparisons without Olaparib treatment, one would think that highlighted common genesets resulted from the treatment effect of Olaparib, not the synergistical interaction between Usp15 and Scaf1.

Line 756, Figure 5 title, "Scaf1 regulates several pathways involved in PDAC development and Olaparib response". This is a strong statement and I do not see direct evidence supporting the statement.

Reviewer #3:

Remarks to the Author:

In this manuscript, Martinez et al. investigate the alternative tumor suppressors in pancreatic ductal adenocarcinoma (PDAC) using an in vivo CRISPR screen. After optimizing their transduction conditions, they screened a library of 125 genes commonly found to be altered in pancreatic cancer patients. From this screen, they focused on the characterization of two hits, USP15 and SCAF1. They demonstrate that both USP15 and SCAF1 have tumor suppressor potential, as individual knockdown of these genes dramatically reduces the survival of KC mice. Functionally, they suggest that USP15 and SCAF1 are both essential for repair of DNA damage induced by PARP inhibition, suggesting a potential clinical opportunity in patients with these mutations.

Overall, the manuscript is well written, and the concepts would be of great interest to the readers of Nature Communications. However, there are areas that the manuscript that could be improved prior to publication.

Major Concerns:

1. There is a lot of reliance on gene signatures to explain reprogramming, but functional validation would be far better. If there is more active NRF2 signaling, are the cells more resistant to redox stress? Is there a measurable difference in mitochondrial metabolism? Are the cells more or less capable of survival in hypoxia as a result? Does the changes in TGF β signaling impact cell migration? While the characterization of all of these is unnecessary, it would be good to see a few of the gene signatures validated.
2. A small but significant number of PDAC patients receive PARP inhibitor treatment. Is there a way the authors can potentially link patient response to the loss of either of their putative tumor suppressors as these are apparently fairly common in patients? I understand these data may not be readily available, but if they can be obtained it would add significant strength to the potential of screening patients for USP15 or SCAF1 to inform treatment.
3. In a similar vein, a significantly higher proportion of PDAC patients that are treated with gemcitabine have been sequenced, it would be useful to potentially mine this data as well to correlate USP15 or SCAF1 to the response.
4. Establishment of 2D KC cell cultures is known to induce loss of oncogene-induced senescence, potentially through loss of p53 function. As such, I caution against making the KC vs. KPC comparison in culture without demonstrating that there is no additional tumor suppressor loss in the KC cells, which would be laborious to show. As the data already exists, I would suggest just treating it as another cell line model vs. drawing a conclusion on the p53 function between the cells. This might be accounted for, but as mentioned in the minor concerns, was not very clear from the sparse methods.
5. Do the authors believe USP15 and SCAF1 mutations are drivers in PDAC alone, or potentially present in other (especially Kras-driven) cancers? If feasible, it might be worth checking in sequence libraries of different cancers, and/or adding to the discussion.

Minor concerns:

1. I believe the duration of the survival curve in Fig 2e is supposed to be in weeks, not days.
2. The materials and methods included in the manuscript are sparse, and as such, it is hard to comment on many of the assays run, such as the cell viability assays. This needs to be corrected on revision.

Reviewer #1 - In vivo CRISPR screens, pancreatic cancer (Remarks to the Author):

In this manuscript, Martinez et al. conducted in vivo CRISPR/Cas9 screens in order to identify novel pancreatic tumor suppressors. Within a library targeting the recurrently mutated pancreatic cancer genes, authors identified USP15 and SCAF1 as tumor suppressor genes. The authors then validated the tumor suppressor function of USP15 in vitro and in vivo conditions and showed that USP15 regulates several immune-related pathways involved in PDAC development. They also provided evidence that the loss of SCAF1 results in the formation of a truncated USP15 isoform. The authors also showed that loss of USP15 or SCAF1 sensitizes cells to PARP inhibition and Gemcitabine.

Overall, the manuscript recommended a mechanism important for PDAC development and therapy response. **One of the study's major strengths is conducting the screenings in vivo, which allowed the authors to identify targets under physiologically relevant conditions.** The **manuscript is well-written** except for a few minor errors stated below. There are a few points that could further enhance conclusions and improve the novelty of the findings in this manuscript forward.

- Authors showed the role of USP15 and SCAF1 alterations, however, they did not provide the actual patient data to support their findings. By analyzing the PDAC cohorts and stratifying the patients based on their USP15 and SCAF1 mutation status (loss of function or gain of function), they can show if there is a correlation with the survival. Since it has been stated that these alterations are observed in 31% of the patients, they would potentially have enough sample size for this analysis.

We thank the reviewer for this comment, and agree that this analysis was largely missing in the original submission. During the revisions, we have focused our work on human PDAC and are happy to report that we are now not only providing novel data from human PDAC samples that further support our findings, but have also been able to obtain several lines of functional evidence showing that USP15 and SCAF1 are indeed tumor suppressor in human PDAC cell lines as well as primary PDAC organoids. This new data is shown in the new **Figure 6** as well as in the **new Supplemental Data Figure 8 and 9** of the revised manuscript:

'To extend our findings from mouse to human cancers, we analysed 295 PDAC samples from The Cancer Genome Atlas⁴⁵⁻⁴⁷. Mutations and homozygous deletion of USP15 and Scaf1 are rare as expected for long-tail mutation and were found in only 2.4% and 1.4% of PDAC samples, respectively. However, an additional 25% and 13% of PDAC cases showed shallow deletions of USP15 and SCAF1, respectively, indicative of heterozygous loss of these genes (please see new Fig. 6a).

Focal *USP15* and *Scaf1* copy-number losses have also been identified in independent large-scale genome studies^{48,49}. In addition, allelic copy number loss coincided with reduced expression of *USP15* and *SCAF1* and patients with deep or shallow *USP15* or *SCAF1* deletions showed a significant trend towards a shorter overall survival (please see new **Fig. 6b** and **Supplementary Data Fig. 8a**):

Supplementary Data Fig. 8a:

Fig. 6b:

Given our genetic and biochemical data linking *SCAF1* and *USP15*, we next considered patients with deep or shallow *USP15* or *SCAF1* deletions as a group (=37% of patients) and found a significant shorter overall survival (please see new **Supplementary Data Fig. 8b**):

This raises the possibility that USP15 and potentially also SCAF1 function in a haploinsufficient manner, which is in line with the increased tumorigenesis found in the *Usp15^{fl/+}*; *KRas^{G12D}*; *Pdx1-Cre* mice.

Next, we assessed expression of USP15 in 4 human pancreatic cancer cell lines. While PANC1 and HPAFII exhibited expression of the small as well as the long USP15 isoform, MiaPACA2 and BXPC3 cell only exhibited low level expression of the long USP15 isoform, indicating that USP15 is also downregulated in some human pancreatic cancer cell lines (**Supplementary Data Fig. 8c**).

To functionally test USP15 and SCAF1, we genetically ablated these genes in human PANC1 cells (**Supplementary Data Fig. 8d and e**). Importantly, genetic ablation of SCAF1 resulted in increased expression of the short USP15 isoform, indicating that this mechanism is conserved from mouse to human cells (**Supplementary Data Fig. 8f**).

Similarly, to our autochthonous mouse experiments, we also found that loss of USP15 or SCAF1 in PANC1 cells resulted in accelerated tumorigenesis and increased sensitivity to olaparib and gemcitabine (**Fig. 6c, d and Supplementary Data Fig. 8g**). We also observed increased NRF2 protein levels in USP15 knockout PANC1 cells, which showed further elevated upon inhibition of TXNRD1/2 and antioxidant imbalance by auranofin treatment⁵⁰ (**Supplementary Data Fig. 8h**), akin to our findings in mouse KC cells. USP15 knockout PANC1 cells also exhibited increased sensitivity to auranofin treatment (**Supplementary Data Fig. 8i**).

Fig. 6c:

Fig. 6d:

Suppl. Fig. 8:

Lastly, we genetically ablated USP15 in patient-derived organoids (PDOs) from 3 different pancreatic cancer patients using Cas9 ribonucleotide particles. We set up competitive growth assays to assess the relative fitness of USP15 knockout PDOs compared to OR2W5 knockout PDOs. Of note, the OR2W5 olfactory receptor is not expressed in pancreatic PDOs and thus serves as control. We mixed the

USP15 knockout and the OR2W5 knockout PDOs at a 1:4 ratio and followed their relative growth by quantifying the percent of USP15 and OR2W5 mutations over time using Sanger sequencing. Within ~10 passages, we observed that the PDO cultures were almost completely taken over by USP15 knockout cells (Fig. 6e). Together, these data demonstrate the tumor suppressive function of USP15 and SCAF1 in pancreatic cancer by modulating several important signalling pathways and that loss of USP15 and SCAF1 sensitizes to gemcitabine and olaparib.'

We believe that these data are very nicely complementing our genetic mouse data and convincingly show that USP15 and SCAF1 also function as tumor suppressors in human PDAC.

Fig. 6e:

- Authors defined the “long-tail” as recurrent but less frequent alterations. They wanted to identify the tumor suppression relevant roles of this long-tail PDAC genes. However, alteration of the genes identified is represented in 31% of the patient population. Since this is a considerable percentage, authors should more clearly explain the cut-off for mutation frequency.

We apologize that this was not sufficiently clear in the submitted first version of the manuscript. The long-tail is based on mutations and is defined by gene mutated at low frequency – usually any genes that is mutated in less than 10% of a given cancer. The 31% of pancreatic cancer patients displaying alterations in USP15 and SCAF1 involves not only mutations but also copy number alterations such as shallow deletions indicative of heterozygous loss. Shallow deletions are relevant in this case as we functionally showed that loss of one USP15 copy accelerates PDAC development, highlighting a role of USP15 as an haploinsufficient tumor suppressor.

We have now clarified this in the new section on human PDAC within the revised manuscript:

‘To extend our findings from mouse to human cancers, we analysed 295 PDAC samples from The Cancer Genome Atlas⁴⁵⁻⁴⁷. Mutations and homozygous deletion of USP15 and Scaf1 are rare as expected for long-tail mutation and were found in only 2.4% and 1.4% of PDAC samples, respectively. However, an additional 25% and 13% of PDAC cases showed shallow deletions of USP15 and SCAF1, respectively,

indicative of heterozygous loss of these genes (Fig. 6a). Focal USP15 and SCAF1 copy-number losses have been identified in independent large-scale genome studies^{48,49}.

- Authors showed that loss of SCAF1 results in the formation of a truncated USP15 isoform and provided rescue experiments where they combined sgScaf1 with USP15 long isoform in vitro. Since this is one of the most important findings of the manuscript, and the hit genes are identified from an in vivo screening set-up, authors should also provide in vivo evidence and conduct similar rescue experiments in vivo.

We agree with this comment and have performed the suggested experiment. We first used CRISPR to knock-out *Usp15* and *Scaf1* in primary Pdx1-Cre; *Kras*^{G12D}; Cas9 cells. These *Usp15* and *Scaf1* knock-out KC cells formed allograft tumor faster than non-targeting control cells (Fig. 2g and 4d).

Interestingly, we found that also *in vivo* SCAF1 KO tumors show slightly less expression of full length USP15 (Supplementary Data Fig. 6g). However, this is caveated by the fact that we are analysing whole tumor lysates and some of the signal also comes from the stroma.

Importantly, re-expressing the full-length isoform of USP15 in *Scaf1* KO cells rescued increased tumorigenesis of *Scaf1* knock-out cells (please, see new Fig. 4h):

- The connection between the depletion of SCAF1 and the depletion of the long form of

USP15 is interesting and warrants further investigation. At a minimum, the connection should be further delineated with rescue experiments. For example, will overexpression of USP15 in SCAF1-depleted cells rescue the SCAF1 depletion phenotype?

We thank this reviewer for this suggestion. We kindly point this reviewer to Fig. 4g of the original manuscript, where we have already conducted this experiment.

In the revised version, we have now also conducted said rescue experiments *in vivo*, highlighted in the previous point and in **new Fig. 4h**:

• Are these results (the connection between SCAF1 and USP15) supported by the patient data? If they sorted TCGA PDAC patients into SCAF1 high and low, will this be correlated with the USP15 long and short isoforms?

Unfortunately, we cannot perform said experiment as there is no data about expression of USP15 protein isoforms available in the TCGA PDAC data. However, we now have conducted functional experiments in human cells and genetic ablation of Scaf1 in the human Panc1 cell line results in increased expression of the short USP15 isoform. (please, see **new Supplementary Data Fig. 8f**):

Importantly and as pointed out before, USP15 and SCAF1 knock-out Panc1 cells formed allograft tumor faster than non-targeting control cells (please, see **new Fig. 6c**):

Secondly, we used CRISPR/Cas9 mediated gene editing in PDAC patient-derived organoids and could show that genetic ablation of USP15 leads to increased proliferation in a competitive growth assay cumulating in outcompeting of USP15wt cells within the pancreatic cancer organoid cultures (please, see **new Fig. 6e**):

Together, these new results further support Usp15 and Scaf1 functioning as suppressors of human pancreatic cancer.

- If depletion of SCAF1 leads to deletion of USP15 long isoform, one might expect a mutual

exclusivity in terms of loss of function mutations between these two genes. Is this true in human PDAC data?

This is an interesting point and we have analysed mutual exclusivity between SCAF1 and USP15 and interestingly we found a tendency towards mutual exclusivity:

In the Witkiewicz et al, 2015 Nature Communications paper, there is even a significant mutual exclusivity between LOF USP15 and SCAF1 mutations:

However, mutual exclusivity is neither a pre-requisite nor a proof that genes work within the same pathway. For example, the p53 pathway (e.g. p53 loss and MDM2/4 amplification) shows really nice mutual exclusivity in GBM:

However, this is not the case in HNSCC:

In fact, deleterious TP53 mutations significantly co-occur with MDM2 amplification. So, looking at the co-occurrence of these p53 pathway genes within HNSCC, one would think that p53, MDM2 and MDM4 are not in the same pathway, which is obviously misleading. This is mainly routed in the fact that inactivation of the p53 pathway is an early step in GBM etiology (= often truncal) but a late step in HNSCC development and different clones within the same tumor inactivate the p53 pathway by different means.

So, while the USP15 and Scaf1 data showing mutual exclusivity in TCGA and Witkiewicz et al, we always view these data with caution as these data sets are relatively small and somewhat underpowered for strong statistical conclusion. Thus, we prefer not to include these data into the manuscript at this point, as it could be over interpreted by some readers. As more data come available,

we will certainly re-visit this analysis and include into future publications. However, if this reviewer and the editor think it would be beneficial, we are happy to include it into the final manuscript.

- Authors showed that loss of USP15 or SCAF1 sensitizes cells to PARP inhibition and Gemcitabine *in vitro*; however, showing that this sensitization works in tumor models would be an important experiment as well.

As suggested by this reviewer, we have now performed these experiments and are happy to report that loss of USP15 or Scaf1 also sensitized allograft tumors *in vivo* towards Olaparib treatment (please see new **Supplementary Data Fig. 4c and 6d**):

- In Supplementary Figure 5e, authors showed the presence of a short USP15 isoform in human Panc1 cells and provided this as evidence for this isoform to be evolutionary conserved. Authors should validate that this isoform is present in a panel of human pancreatic cancer cell lines to support this data.

We have now assessed expression of USP15 in 4 human pancreatic cancer cell lines. While PANC1 and HPAFII exhibited expression of the small as well as the long USP15 isoform, MiaPACA2 and BxPC3 cell only exhibited low level expression of the long USP15 isoform, indicating that USP15 is also downregulated in some human pancreatic cancer cell lines (please see new **Supplementary Data Fig. 8c**):

C

In addition, we now also show WB data that genetic ablation of SCAF1 in PANC1 cells leads to increase expression level of the short USP15 isoform, indicating that the isoform but also its regulation by SCAF1 is conserved in human cells (please see new **Supplementary Data Fig. 8f**):

- How does the sgRNA detection frequency of each gene in vivo correlate with the mutation frequency of the target gene in TCGA data? Can it be concluded that the most frequent genes are also the ones that can form tumors most frequently?

Looking at our hits, in general, we do not find such a correlation. For example, while *Usp15* is our second most common hit in our mouse screen, *USP15* is mutated in only 2.4% of PDAC patients. However, we found a substantial proportion of shallow *USP15* deletions and our genetic mouse experiments could show that *USP15* functions in a haploinsufficient manner (please see **Fig. 6a**). Together, this indicates that some of the less well-known or hitherto unknown tumor suppressor genes might be mutated at low frequency, but deleted or otherwise inactivated in a substantially larger fraction of patients. Indeed, we have observed very similar patterns in our recent efforts to characterize driver genes in Head and Neck cancer (Loganathan et al, Science 2020) as well as breast cancer (Langille et al., Cancer Discovery 2022) or lung cancer (Dervovic et al., Nature Comm 2023).

As such, when one includes copy-number alterations, there is a nice correlation between the effect in our mouse screen and the human data. However, as copy number alterations are so prevalent and encompass hundreds to thousands of genes, prior functional knowledge is needed to make such correlations. In conclusion, we actually think that the multiplexed functional characterization of driver genes is one of the most important contribution of this manuscript.

- Finally, the Authors never talked about the Mets in liver and lungs. The model systems look like a great platform to discover if certain sgRNAs are selectively enriched in specific metastasis locations. Authors should analyze metastatic lesions and present findings about the metastatic potential of sgRNAs targeting various genes.

In the original submitted manuscript, we have shown that 56% KC mice transduced with the long-tail PDAC sgRNA library developed liver and/or lung metastasis, while only 8% littermate mice transduced

with the control sgRNA library developed metastasis (please see original Supplementary Data Fig. 2c), showing that the long-tail library significantly increased the metastatic behaviour of pancreatic cancer.

However, while we could track and numbered those metastases using our H2B-RFP system, most of them were rather small and difficult to sequence and most mice had to be sacrificed due to the primary tumor burden before the metastatic foci could grow out. We managed to only sequence ~20% of the metastatic lesions, and those showed a good correlation between the enrichment sgRNA in the primary tumor and the metastasis (please, see **new Supplementary Data Fig. 2d** and **new sheet in Supplementary table 2**), with Fbxw7, USP15, Scaf1, Cdkn2a and Rnf43 as top hits.

Minor points:

- In the text body, Figure 2e is labeled as Figure 2c, and Supplementary Figure 3f is labeled as Supplementary Figure 3e.

Thank you for pointing this out. We are sorry for these oversights and have corrected these mistakes in the revised manuscript.

Reviewer #2 - Computational biology (Remarks to the Author):

Lines 62-63: “USP15 and SCAF1 alterations are observed in 31% of pancreatic cancer patients”. Is this true? Please provide a reference.

We apologize that this was not sufficiently clear in the submitted first version of the manuscript. The long-tail is based on mutations and is defined by gene mutated at low frequency – usually any gene mutated in less than 10% of a given cancer. The 31% of pancreatic cancer patients displaying alterations in USP15 and SCAF1 involves not only mutations but also copy number alterations such as shallow deletions indicative of heterozygous loss. Shallow deletions are relevant in this case as we functionally showed that loss of one USP15 copy accelerates PDAC development, highlighting a role of USP15 as an haploinsufficient tumor suppressor.

We have now clarified this in the new section on human PDAC within the revised manuscript:

‘First, we have analysed 295 PDAC samples from The Cancer Genome Atlas, which revealed that mutations and homozygous deletion of USP15 and Scaf1 are rare as expected for long-tail mutations and were found in only 2.4% and 1.4% of PDAC samples, respectively. However, an additional 25% and 13% of PDAC cases showed shallow deletions of USP15 and Scaf1, respectively, indicative of frequent heterozygous loss of these genes (please see new Fig. 6a). Focal USP15 and Scaf1 copy-number losses have also been identified in independent large-scale genome studies^{48,49}.’

Line 165: “further supporting their functions as strong suppressors of pancreatic cancer”. This is a strong statement based on one mouse model (KC). Perhaps rephrase to “further supporting their functions as strong suppressors of pancreatic cancer in KC mice”?

We agree with this reviewer and have changed the text as suggested.

In addition, to also assess the role of USP15 and SCAF1 in humans, we focused much of our revision on human PDAC and are happy to report that we now not only providing novel data from human PDAC samples that further support our findings, but have also been able to obtain several lines of functional evidence showing that USP15 and SCAF1 are indeed tumor suppressor in human PDAC cell lines as well as primary PDAC organoids. This new data is shown in the new **Figure 6** as well as in the **new Supplemental Data Figure 8 and 9** of the revised manuscript:

‘To extend our findings from mouse to human cancers, we analysed 295 PDAC samples from The Cancer Genome Atlas⁴⁵⁻⁴⁷. Mutations and homozygous deletion of USP15 and Scaf1 are rare as expected for long-tail mutation and were found in only 2.4% and 1.4% of PDAC samples, respectively. However, an additional 25% and 13% of PDAC cases showed shallow deletions of USP15 and SCAF1, respectively, indicative of heterozygous loss of these genes (Fig. 6a). Focal USP15 and SCAF1 copy-number losses have been identified in independent large-scale genome studies^{48,49}. In addition, allelic copy number loss

also coincided with reduced expression of USP15 and SCAF1 and patients with deep or shallow USP15 or SCAF1 deletions showed a significant trend towards a shorter overall survival (**Fig. 6b and Supplementary Data Fig. 8a**). Given our genetic and biochemical data linking SCAF1 and USP15, we next considered patients with deep or shallow USP15 or SCAF1 deletions as a group (=37% of patients) and found a significant shorter overall survival (**Supplementary Data Fig. 8b**). This raises the possibility that USP15 and potentially also SCAF1 function in a haploinsufficient manner, which is in line with the increased tumorigenesis found in the *Usp15^{fl/+}; KRas^{G12D}; Pdx1-Cre* mice.

Next, we assessed expression of USP15 in 4 human pancreatic cancer cell lines. While PANC1 and HPAFII exhibited expression of the small as well as the long USP15 isoform, MiaPACA2 and BXPC3 cell only exhibited low level expression of the long USP15 isoform, indicating that USP15 is also downregulated in some human pancreatic cancer cell lines (**Supplementary Data Fig. 8c**).

To functionally test USP15 and SCAF1, we genetically ablated these genes in human PANC1 cells (**Supplementary Data Fig. 8d and e**). Importantly, genetic ablation of SCAF1 resulted in increased expression of the short USP15 isoform, indicating that this mechanism is conserved from mouse to human cells (**Supplementary Data Fig. 8f**). Similarly, to our autochthonous mouse experiments, we also found that loss of USP15 or SCAF1 in PANC1 cells resulted in accelerated tumorigenesis and increased sensitivity to olaparib and gemcitabine (**Fig. 6c, d and Supplementary Data Fig. 8g**). We also observed increased NRF2 protein levels in USP15 knockout PANC1 cells, which showed further elevated upon inhibition of TXNRD1/2 and antioxidant imbalance by auranofin treatment⁵⁰ (**Supplementary Data Fig. 8h**), akin to our findings in mouse KC cells. USP15 knockout PANC1 cells also exhibited increased sensitivity to auranofin treatment (**Supplementary Data Fig. 8i**).

Lastly, we genetically ablated USP15 in patient-derived organoids (PDOs) from 3 different pancreatic cancer patients using Cas9 ribonucleotide particles. We set up competitive growth assays to assess the relative fitness of USP15 knockout PDOs compared to OR2W5 knockout PDOs. Of note, the OR2W5 olfactory receptor is not expressed in pancreatic PDOs and thus serves as control. We mixed the USP15 knockout and the OR2W5 knockout PDOs at a 1:4 ratio and followed their relative growth by quantifying the percent of USP15 and OR2W5 mutations over time using Sanger sequencing. Within ~10 passages, we observed that the PDO cultures were almost completely taken over by USP15 knockout cells (**Fig. 6e**). Together, these data demonstrate the tumor suppressive function of USP15 and SCAF1 in pancreatic cancer by modulating several important signalling pathways and that loss of USP15 and SCAF1 sensitizes to gemcitabine and olaparib.'

We believe that these data are very nicely complementing our genetic mouse data and convincingly show that USP15 and SCAF1 also function as tumor suppressors in human PDAC.

Line 216, "(FDR)<0.05", please specify the FDR method used.

We used the Benjamini-Hochberg procedure built into the deseq2 package.

Line 219-220, "These findings are in line with USP15's known role in negatively regulating NRF2 (encoded by the NFE2L2 gene)", please provide a reference.

We are sorry for this oversight and have added the reference in the revised manuscript. Thank you for pointing this out.

Line 224, “GSEA also revealed decreased gene sets”. Do you mean “depleted gene sets”? Were those also up-regulated genes?

Thank you for pointing this out – we have changed the wording to ‘depleted’. The up-regulated gene sets are discussed in the preceding paragraph.

Line 514: please specify the cutoff used for determining enriched gene sets (NES cutoff)
The NES cutoff used is +/- 1.4.

Figure 3C, I am puzzled by the heatmap bars (NES). Enrichment scores only apply to gene sets, not individual genes. What is shown in the heatmaps – gene expression levels or normalized enrichment score (NES)?

We are sorry for this oversight. Heatmap displays gene expression level. This mistake in the figure and legend has been corrected.

Also, in Figure 3C, the authors showed the enrichment plots for two pathways. The most significant pathway with deleted gene set – the cytokine receptor binding – is not shown. Conversely, none of significant pathways with enriched gene sets were shown. I am curious about how these choices were made.

We decided to show TNFa signaling because we later functionally tested the TNFa signaling and confirmed reduced TNFa signaling in USP15 knockout cells. We now show the enrichment plots of the two most significant pathways with enriched gene sets in Suppl. Data Figure 5b. TNFa signaling via NFKB is the most significant deleted gene set, this mistake in the Figure 3C bar graph has been corrected.

Figure 5C, genes in glycolysis pathway were significantly enriched in both sgUsp15 vs sgCtrl+Olaparib and sgScaf1 vs sgCtrl+Olaparib comparisons. The authors highlighted pathways with enriched/depleted genesets common to both comparisons. Since little in common was found between the comparisons without Olaparib treatment, one would think that highlighted common genesets resulted from the treatment effect of Olaparib, not the synergistical interaction between Usp15 and Scaf1.

We apologize if this experimental set-up was not explained clearly enough. We compared the sgUsp15 + **Olaparib** versus sgCTRL + Olaparib as well as sgScaf1 + **Olaparib** versus sgCTRL + Olaparib and as such, it is unlikely that the ‘*common genesets resulted from the treatment effect of Olaparib*’ but that the common genesets are a result of the individual genetic perturbation of Usp15 and Scaf1. In addition, we actually thought that it is quite remarkable that 2 out of the 4 most upregulated and 4 out of the 6 most downregulated pathways in Usp15 KO cells treated with Olaparib also are the most up/down-regulated pathways in the Scaf1 Ko cells treated with Olaparib when compared to Olaparib-treated control cells. In addition, we validated this finding in an intendent experiment using RT-PCR for Hedgehog induced genes.

In the revised version of the manuscript, we now clarified in the labels in the figure and replaced sgUsp15 versus sgCTRL + Olaparib with sgUsp15 + **Olaparib** versus sgCTRL + Olaparib and sgScaf1 versus sgCTRL + Olaparib with sgScaf1 + **Olaparib** versus sgCTRL + Olaparib (please see Fig. 5C)

Line 756, Figure 5 title, “Scaf1 regulates several pathways involved in PDAC development and Olaparib response”. This is a strong statement and I do not see direct evidence supporting the statement.

We agree with this reviewer and have re-phrased this figure title to: “Scaf1 regulates TNF α and p53 signaling as well as hedgehog signaling in response to Olaparib”.

To further substantiate this statement, we performed further qRT-PCR, which indeed revealed dysregulated TNF α signaling (please, see **new Supplementary Data Fig. 7g**):

G

Reviewer #3 - Pancreatic cancer (Remarks to the Author):

In this manuscript, Martinez et al. investigate the alternative tumor suppressors in pancreatic ductal adenocarcinoma (PDAC) using an in vivo CRISPR screen. After optimizing their transduction conditions, they screened a library of 125 genes commonly found to be altered in pancreatic cancer patients. From this screen, they focused on the characterization of two hits, USP15 and SCAF1. They demonstrate that both USP15 and SCAF1 have tumor suppressor potential, as individual knockdown of these genes dramatically reduces the survival of KC mice. Functionally, they suggest that USP15 and SCAF1 are both essential for repair of DNA damage induced by PARP inhibition, suggesting a potential clinical opportunity in patients with these mutations.

Overall, the **manuscript is well written, and the concepts would be of great interest to the readers of Nature Communications**. However, there are areas that the manuscript that could be improved prior to publication.

Major Concerns:

1. There is a lot of reliance on gene signatures to explain reprogramming, but functional validation would be far better. If there is more active NRF2 signaling, are the cells more resistant to redox stress? Is there a measurable difference in mitochondrial metabolism? Are the cells more or less capable of survival in hypoxia as a result? Does the changes in TGF β signaling impact cell migration? While the characterization of all of these is unnecessary, it would be good to see a few of the gene signatures validated.

We agree with this reviewer and have added several lines of functional validation in the revised manuscript: 1, As suggested, we first evaluated increased NRF2 signaling in PANC1 cells. First, we knocked out USP15 in these human PDAC cells, which resulted in increased NRF2 levels, which is in line with the findings from the genetic mouse tumor. In addition, we observed further elevated of NRF2 levels upon inhibition of TXNRD1/2 and antioxidant imbalance by auranofin treatment⁵⁰ (please see new **Supplementary Data Fig. 8h**) and USP15 knockout PANC1 cells also exhibited increased sensitivity to auranofin treatment (please see new **Supplementary Data Fig. 8i**). Together, this clearly shows that USP15 is regulating NRF2 signalling and the ability of cells to survive redox stress:

Supplementary Data Fig. 8h and i:

2, As suggested, we also tested TGFβ-induced migration and USP15 not only regulated expression of TGFβ-responsive genes, but loss of USP15 also impairs TGFβ-induced migration (please see new **Supplementary Data Fig. 5e**):

3, In addition, we evaluated whether USP15 regulates TNFα-induced cell death. Indeed, loss of USP15 not only reduced expression of TNFα-target genes, but also leads to reduced TNFα-induced cell death (please see new **Supplementary Data Fig. 5d**):

In addition, we have validated the signatures for TNFα and TGFβ signaling in Usp15 knock-out cells using RT-PCR for target gene expression (please see Fig. 3d and Supplemental Data Fig. 5c), TNFα signalling in Scaf1 knock-out cells (please see new **Supplemental Data Fig. 7g**) as well as for hedgehog signaling under PARPi in Usp15 and Scaf1 knock-out cells (please see Fig. 5d).

2. A small but significant number of PDAC patients receive PARP inhibitor treatment. Is there a way the authors can potentially link patient response to the loss of either of their putative tumor suppressors as these are apparently fairly common in patients? I understand these data may not be readily available, but if they can be obtained it would add significant strength to the potential of screening patients for USP15 or SCAF1 to inform treatment.

This is indeed a very interesting and potentially clinically important point. Unfortunately, PARPi response Data were not available in our COMP251 cohort, which precluded this analysis. However, this will be the focus for future follow-up studies.

3. In a similar vein, a significantly higher proportion of PDAC patients that are treated with gemcitabine have been sequenced, it would be useful to potentially mine this data as well to correlate USP15 or SCAF1 to the response.

This is an interesting aspect and while there is no reliable data for the TCGA cohorts, we collaborated with the Toronto PanCURX team led by Drs. Steve Gallinger and Fayiaz Notta and analysed their data. Interestingly, we observed that tumors with high SCAF1 expression were enriched in gemcitabine non-responders (PD). While this is in line with our functional data showing that loss of SCAF1 sensitizes cells towards gemcitabine treatment, this effect in the relatively small human cohort was just a trend:

To further explore this observation, we next stratified the top vs. bottom 10% of SCAF1 expressing tumors and observed a significant enrichment of SCAF1 high expressing tumor in gemcitabine non-responders (PD).

While this data is encouraging, we failed to observe similar trends when stratifying USP15 expression. This can have manifold reasons, esp. because gemcitabine (as well as Olaparib) is itself upregulating

USP15 (please see Supplementary Data Fig. 4c) and this happens presumably not only the tumor cells but also in the stroma and as such can be a confounding factor in this analysis.

While the SCAF1 data is certainly supporting our findings, we feel that overall, this needs much deeper exploration. It is also important to note that interpretation of all these results has to be done with caution, as they can be confounded by many different variables (such as USP15 expression due to gemcitabine treatment) that are not controlled for in these analyses. In addition, it is important to note that all the patients have received treatment, and it is likely that USP15 will also sensitize to other treatments such as Olaparib (as shown in the manuscript) or FOLFIRINOX. As such, we prefer not to include these data into the manuscript at this point and want to further study the effects of USP15 and SCAF1 in PDAC treatment in our follow-up studies.

However, to extend and corroborate our findings in human PDAC, we have now added considerable new data and added a whole new figure 6 and supplemental fig 8 and 9 to the paper, showing correlative as well as functional data confirming the tumor suppressive function of USP15 and SCAF1 in human pancreatic cancer.

4. Establishment of 2D KC cell cultures is known to induce loss of oncogene-induced senescence, potentially through loss of p53 function. As such, I caution against making the KC vs. KPC comparison in culture without demonstrating that there is no additional tumor suppressor loss in the KC cells, which would be laborious to show. As the data already exists, I would suggest just treating it as another cell line model vs. drawing a conclusion on the p53 function between the cells. This might be accounted for, but as mentioned in the minor concerns, was not very clear from the sparse methods.

We agree with this reviewer that oncogene-induced senescence, potentially through loss of p53 function or loss of other tumor suppressor genes is a potential caveat. To explore this potential caveat, we first tested the p53 response of the KC cells using nutlin treatment, which efficiently triggered upregulation of p53 protein and p53 target genes such as p21 and MDM2, indicating an intact p53 response in KC cells (please see Suppl 3g):

In line with this reviewer comment, it is interesting to note that the primary KC cells only proliferate for about 15 passage in culture before they senesce and detach from the plate.

In addition, we also agree with this reviewer that adding additional model systems to confirm the function of USP15 and SCAF1 would be beneficial. We have now used the human a Panc1 Cas9 cell line and used CRISPR to knock-out USP15 and SCAF1. Of note, USP15 and SCAF1 knock-out Panc1 cells formed allograft tumor faster than non-targeting control cells (please, see **new Fig. 6c**):

Secondly, we used CRISPR/Cas9 mediated gene editing in PDAC patient-derived organoids and could show that genetic ablation of USP15 leads to increased proliferation in a competitive growth assay cumulating in outcompeting of USP15wt cells within the pancreatic cancer organoid (please, see **new Fig. 6e**):

Together, these new results further support *Usp15* and *Scaf1* function as suppressors of pancreatic cancer.

5. Do the authors believe USP15 and SCAF1 mutations are drivers in PDAC alone, or potentially present in other (especially *Kras*-driven) cancers? If feasible, it might be worth checking in sequence libraries of different cancers, and/or adding to the discussion.

We don't think that USP15 or SCAF1 mutations alone are drivers in PDAC. We have now aged mice *USP15^{fl/fl}*, *Pdx1-Cre* mice for over 1.5 years but did not observe pancreatic cancer development.

Regarding other cancer types, we think this is indeed an interesting aspect, which we now address in the discussion and in the new Supplementary Data Fig. 9a:

'For example, it will be interesting to see whether USP15 functions as haploinsufficient tumor suppressor also in other cancers that show frequent shallow USP15 deletion such as sarcoma, esophageal adenocarcinoma, melanoma or lung cancers.'

Minor concerns:

1. I believe the duration of the survival curve in Fig 2e is supposed to be in weeks, not days.

Figure 2e shows an Incucyte experiment where growth of KC cells transduced with the indicated sgRNAs are measured over 5 days using cell confluency as a proxy. As such, this growth curve is in days.

2. The materials and methods included in the manuscript are sparse, and as such, it is hard to comment on many of the assays run, such as the cell viability assays. This needs to be corrected on revision.

We have revised the methods and added more details.

Reviewers' Comments:

Reviewer #1:

Remarks to the Author:

The authors did a great job addressing all the concerns that were raised. The manuscript has been significantly improved. I do not have further comments.

Reviewer #2:

Remarks to the Author:

The authors have adequately addressed my comments.

Reviewer #3:

Remarks to the Author:

The authors have addressed my previous comments, I have no further concerns. It is widely known that there are major roles for putative oncogenic drivers/tumor suppressors beyond KRAS, p53, SMAD4, and p16 this is a great technique and study to begin to address these.